# RGS2 drives male aggression in mice via the serotonergic system

Melanie D. Mark [1]*, Patric Wollenweber[1], Annika Gesk[1], Katja Kösters[1], Katharina Batzke[1], Claudia Janoschka[1], Takashi Maejima[2], Jing Han[3], Evan S. Deneris[4] & Stefan Herlitze[1]

Aggressive behavior in our modern, civilized society is often counterproductive and destructive. Identifying specific proteins involved in the disease can serve as therapeutic targets for treating aggression. Here, we found that overexpression of RGS2 in explicitly serotonergic neurons augments male aggression in control mice and rescues male aggression in $Rgs2^{-/-}$ mice, while anxiety is not affected. The aggressive behavior is directly correlated to the immediate early gene *c-fos* induction in the dorsal raphe nuclei and ventrolateral part of the ventromedial nucleus hypothalamus, to an increase in spontaneous firing in serotonergic neurons and to a reduction in the modulatory action of $G_{i/o}$ and $G_{q/11}$ coupled 5HT and adrenergic receptors in serotonergic neurons of $Rgs2$-expressing mice. Collectively, these findings specifically identify that RGS2 expression in serotonergic neurons is sufficient to drive male aggression in mice and as a potential therapeutic target for treating aggression.

[1] Department of General Zoology and Neurobiology, Ruhr-University Bochum, 44780 Bochum, Germany. [2] Department of Integrative Neurophysiology, Graduate School of Medical Sciences, Kanazawa University, Kanazawa 920-8640, Japan. [3] Institute for Applied Cancer Science, University of Texas, MD Anderson Cancer Center, Houston, TX 77030, USA. [4] Department of Neurosciences, Case Western Reserve University, Cleveland, OH, USA. *email: Melanie.mark@rub.de

Regulators of G protein signaling (RGS) proteins terminate G protein signals by accelerating the intrinsic GTPase activity of the G protein. In addition, RGS proteins can act as effector antagonists by blocking, for example, $G_{i/o}$, $G_{q/11}$, and $G_s$ pathways[1,2]. The RGS family comprises more than 20 different family members identified in mammalian tissues where RGS2 belongs to the subgroup of small RGS proteins. This subgroup is characterized by small N- and C-termini flanking a conserved 130 amino acid long intra-protein domain also known as RGS domain, which is responsible for their interaction and activity on the G protein α-subunit[3].

RGS2 is expressed in the periphery and brain and has been identified as an immediate early gene (IEG). Rapid upregulation of RGS2 expression has been shown, for example, in various brain regions like the hippocampus by excitatory stimuli[4]. These results suggest a regulatory role of RGS2 in synaptic transmission and synaptic plasticity during behavior. Indeed, RGS2 increases synaptic vesicle release at the presynaptic terminals via down-regulation of $G_{i/o}$-mediated $Ca^{2+}$ channel inhibition and $Rgs2^{-/-}$ mice reveal a decrease in synaptic transmission, but increase in long-term potentiation in the hippocampus[5]. The changes in synaptic plasticity may contribute to the increased anxiety and reduced male aggression of the $Rgs2^{-/-}$ mice[5–7].

RGS2 has been identified as a quantitative trait for anxiety and variations within the $Rgs2$ gene are expected to play a role for the development of anxiety in humans[8,9]. Human genetic association studies linking RGS2 polymorphisms to panic disorder, generalized anxiety disorder, social anxiety disorder, and post-traumatic stress disorder[10–16]. Past meta-analytical integration studies examining the anxiety disorders in twins showed a genetic heritability of 31.6%. Generalized anxiety disorders (GADs) are responsible for 23% genetic variance, with the rest being accountable by unusual environmental factors. Several genetic studies implicate GAD susceptibility genes in the serotonergic and catecholaminergic systems, which are in part modulated by RGS2 (i.e. 5HTT, 5HT1A, MAOA) and the BDNF gene. Additionally, polymorphisms in the neuropeptide S receptor ($NPSR1$), neuropeptide Y receptor ($NPYR$), and corticotropin-releasing hormone receptor ($CRHR1$) genes with individuals exposed to catastrophic events such as a hurricane or early trauma are more susceptible to GAD and anxiety sensitivity. More importantly, RGS2 demonstrated a dose–response correlation to post hurricane GAD in mostly female victims[17,18]. Polymorphisms in specific genes appear to predispose individuals to GAD and anxiety sensitivity under stressful or traumatic environmental conditions.

Anxiety disorder and phenotypes are often associated with altered aggression, suggesting that overlapping and interconnected neuronal circuits are involved in the modulation of both behaviors[19,20]. Mouse models knocking out $Rgs2$ ($Rgs2^{-/-}$) show increased anxiety and decreased male aggression[7,21]. Furthermore, mice treated with oxytocin demonstrated decreased anxiety with increased RGS2 expression[22]. Numerous reports in humans also implicate serotonin (5HT) involvement in regulating impulsive aggression. Functional polymorphisms in humans to the 5HT receptors and 5HT transporter (5HTT) were found to be associated with impulsive aggressive behaviors[23]. In addition, humans with polymorphisms in enzymes, which influence the production, release, and degradation of 5HT, are genetically predisposed to aggression. Polymorphisms identified in another enzyme, tryptophan hydroxylase (TPH) that catalyzes the rate-limiting step in 5HT synthesis, has been associated with susceptibility to borderline personality disorder, aggression, impulsivity, and suicidal behavior[23]. Together, these data suggest that 5HT may be the common neurotransmitter modulating and interconnecting the neuronal circuits underlying anxiety and

aggression[19,24]. This is, for example, suggested by the fact that when the majority of serotonergic neurons lose their 5HT identity in the brain, it leads to increased aggression and anxiety in mice[25]. In general, changes in 5HT levels in the brain have been associated with altered anxiety and aggression. However, how changes in 5HT activity and 5HT release during behavior relate to an increase or decrease in anxiety and aggression is still controversial and may depend on the behavioral context and genetic background of the animal. In addition, whether RGS proteins such as RGS2 modulate anxiety and aggression via modulating the activity of 5HT neurons have not been explored.

$Rgs2$ mRNA has been detected in the dorsal raphe nuclei (DRN), one of the brain nuclei where serotonergic neurons originate[26]. Various $G_{i/o}$- and $G_q$-coupled GPCRs (G protein-coupled receptors), including $5HT_{1A}$ and $5HT_{1B}$ and $α_1$ adrenergic ($α_1AR$) receptors, have been identified in the serotonergic system and recent studies performed in heterologous expression systems showed that these receptor pathways are modulated by RGS proteins including RGS2 (refs[27,28]). Furthermore, RGS2 expression levels seem to correlate with $5HT_{1A}$ and $5HT_{1B}$ receptor expression levels in the DRN[21]. To investigate whether RGS2 regulates anxious and aggressive behavior via the serotonergic system in mice, we generated mice overexpressing $Rgs2$ specifically in serotonergic neurons. Exogenous expression of $Rgs2$ only in serotonergic neurons of the DRN augmented aggressive but not anxious behavior in mice. In addition, $Rgs2$-overexpressing mice were able to rescue the aggressive but not anxious phenotype in $Rgs2^{-/-}$ mice. Interestingly, exogenous expression of $Rgs2$ specifically in serotonergic neurons leads to increases in serotonergic firing, reduction/block of $5HT_{1B}$ and $α_1AR$ modulation, and increased $c$-$fos$ expression in the DRN and ventrolateral area of the ventromedial nucleus hypothalamus (VMHvl) after aggressive behavior.

## Results

**Detection of $Rgs2$ mRNA in serotonergic neurons.** Since the serotonergic system and RGS2 was previously found to be involved in anxiety and aggression in mice[7,25], we wanted to investigate whether RGS2 is modulating anxiety and aggression through the serotonergic system. Since there is no specific antibody available for RGS2 to determine if $Rgs2$ was expressed in serotonergic neurons, we used fluorescence-activated cell sorting (FACS) of YFP-expressing serotonergic neurons[5,29,30], followed by an Affymetrix gene array to detect differences in $Rgs2$ and the closely related RGS family member $Rgs4$ mRNA levels in serotonergic (5HT) neurons. RGS4 has also been described to modulate synaptic plasticity and to be expressed in the DRN in non-serotonergic neurons[30,31]. Dissociated serotonergic neurons from the DRN of E12.5 ePet-YFP$^+$ mice were previously enriched with FACS where serotonergic neurons are expressing YFP (YFP$^+$). $Rgs2$ mRNA was expressed in serotonergic YFP$^+$ neurons but not $Rgs4$ (Fig. 1a). We also confirmed the $Rgs2$ mRNA expression in hippocampal cultures from wild-type mice but not $Rgs2^{-/-}$ mice as a control. 18S RNA was used as an internal control to normalize the relative $Rgs2$ and $Rgs4$ mRNA levels (Fig. 1c). To verify the serotonergic specificity of DRN-dissociated neurons, $Pet$-$1$ and tryptophan hydroxylase 2 ($Tph2$) levels were measured in serotonergic YFP$^+$ and YFP$^-$ neurons from the DRN. mRNA levels of $Pet$-$1$ and $Tph2$ was evident in YFP$^+$ but not YFP$^-$ neurons (Fig. 1b). Low levels of $Pet$-$1$ are not surprising since Pet-1 is a transcription factor initially being expressed at around E10.5. These findings demonstrate that $Rgs2$ is expressed in serotonergic neurons and suggest that Rgs2 has a specific function in the serotonergic transmitter system.

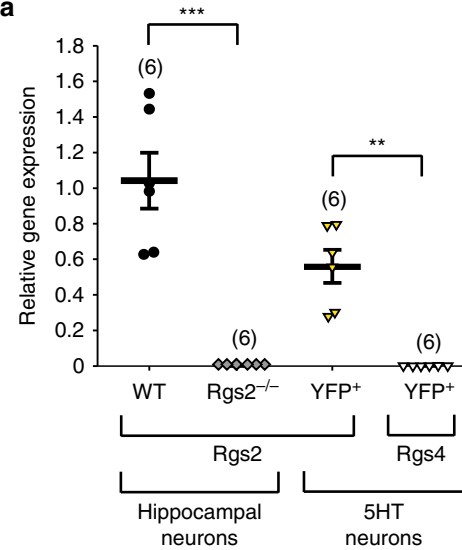

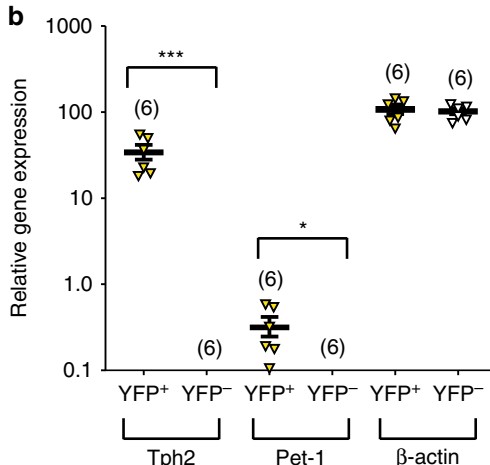

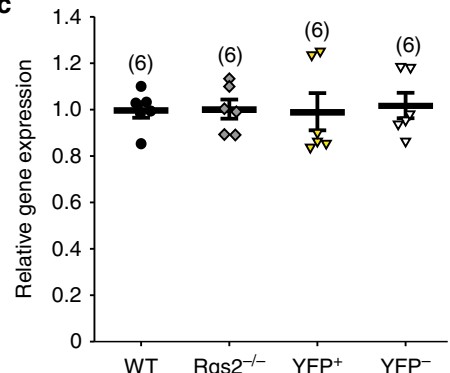

**Fig. 1** *Rgs2* mRNA is detected in YFP-positive, serotonergic neurons by real-time quantitative PCR. **a** The relative *Rgs2* mRNA expression from hippocampal neurons cultured from wild-type (WT; black circles) or *Rgs2*-knockout (*Rgs2⁻/⁻*, grey diamonds) mice. As expected, WT hippocampal neurons express *Rgs2* mRNA but not *Rgs2⁻/⁻* hippocampal neurons. The relative *Rgs2* (yellow triangles) or *Rgs4* (white triangles) mRNA expression from serotonergic neurons cultured from ePet-YFP⁺ (YFP⁺), embryonic day 12.5 (E12.5) mice. mRNA from *Rgs2* but not *Rgs4* was detected. **b** The YFP-positive (YFP⁺) serotonergic neurons but not the YFP-negative neurons (YFP⁻, white triangles) express the *Pet-1* and tryptophan hydroxylase 2 (*Tph2*) showing that YFP⁺ (yellow triangles) neurons are serotonergic neurons. *β-actin* was measured as an internal loading control and expressed at equal levels in the YFP⁺ and YFP⁻ neurons. The low level of *Pet-1* is most likely related to the fact that *Pet-1* is a transcription factor initially expressed at E10.5 when the 5HT cells were sorted. **c** To compare the relative mRNA levels of *Rgs2*, *Rgs4*, *Tph2*, *Pet-1*, and *β-actin* from **a**, **b** in different neuronal cultures 18S RNA was used as an internal control and normalized to the amount of 18S RNA found in hippocampal or serotonergic neurons from each respective mRNA sample. The experiments were performed with three independent neuronal cultures in duplicates ($n = 6$). Relative gene expression data was analyzed with $2$−$\Delta\Delta$CT method[80] and reported as mean ± SD. Statistical significance was evaluated with ANOVA (*$p < 0.05$, **$p < 0.01$, ***$p < 0.001$)

highly reproducible 5HT-specific expression among independent transgenic mouse lines[29,33]. For stable integration of *Rgs2* into the mouse genome, the Rgs2-IRES-GFP construct was subcloned 3′ to the *β-globin* minimal promoter and 40 kb serotonergic-specific *ePet-1* enhancer in the modified pBACe3.6 vector (Fig. 2a) to create BAC transgenic mouse lines expressing *Rgs2* specifically in serotonergic neurons. We selected two of the nine founder lines, which bred well and revealed a low and high expression of RGS2/GFP specifically in 5HT neurons (lines ePet-Rgs2^lo and ePet-Rgs2^hi; Fig. 2c). Quantitative analysis of the GFP intensity in 5HT neurons from ePet-Rgs2^hi compared to ePet-Rgs2^lo mice revealed a 2-fold increase in GFP expression (Fig. 2b). Since no specific antibody is available for RGS2, we analyzed *Rgs2* mRNA levels in serotonergic neurons by single-cell reverse transcriptase quantitative PCR (qPCR) from ePet-Rgs2 transgenic mouse lines crossed with ePet-YFP for identification of serotonergic neurons. *Rgs2* mRNA levels were 20.6% higher in serotonergic neurons from ePet-Rgs2^hi compared to ePet-Rgs2^lo mice (Fig. 2d, e).

**Overexpression of RGS2 in serotonergic neurons leads to male aggression.** The transgenic mouse lines, ePet-Rgs2^lo and ePet-Rgs2^hi, exogenously expressing RGS2 in the 5HT system were characterized for male anxiety, depression, and aggressive behaviors in comparison to their wild-type littermates. Both ePet-Rgs2 transgenic mouse lines demonstrated minimal differences in their anxiolytic or depressive behavior compared to their control littermates in the open field, elevated plus maze, place preference, forced swim, or tail suspension tests (Table 1). However, ePet-Rgs2^hi male mice exhibited augmented aggressive behavior in the resident intruder and tube dominance tests. ePet-Rgs2^hi male mice displayed an increased number and duration of attacks and bites, as well as a shorter latency to their first attack in comparison to their littermate controls and ePet-Rgs2^lo male mice in the resident intruder test (Fig. 3a–d, Supplementary Movies 1 and 2). They also demonstrated an increase in number and duration of attacks in the resident intruder test with pups (Fig. 3e, f). Interestingly, the ePet-Rgs2^lo male mice indicated a trend for more aggressive behavior in the resident intruder test compared to their littermate controls, which may indicate a RGS2 dosage effect (Fig. 3a–d). Furthermore, the tube dominance test revealed that

**Creation of ePet-Rgs2 transgenic mouse lines**. To investigate the physiological function and behavioral consequences of *Rgs2* expression in the serotonergic system of mice, we used the *Pet-1* enhancer region to overexpress *Rgs2* specifically in the serotonergic system. The transcription factor *Pet-1* is required for the development of the serotonergic system in the mouse brain[30,32,33]. A 5′ genomic fragment flanking the mouse *Pet-1* gene restricts the expression of the reporter gene β-galactosidase to serotonergic neurons during development and in adult mice, indicating that the gene of interest *Rgs2* will also be restricted to serotonergic neurons. Most importantly, the *Pet-1* reporter shows

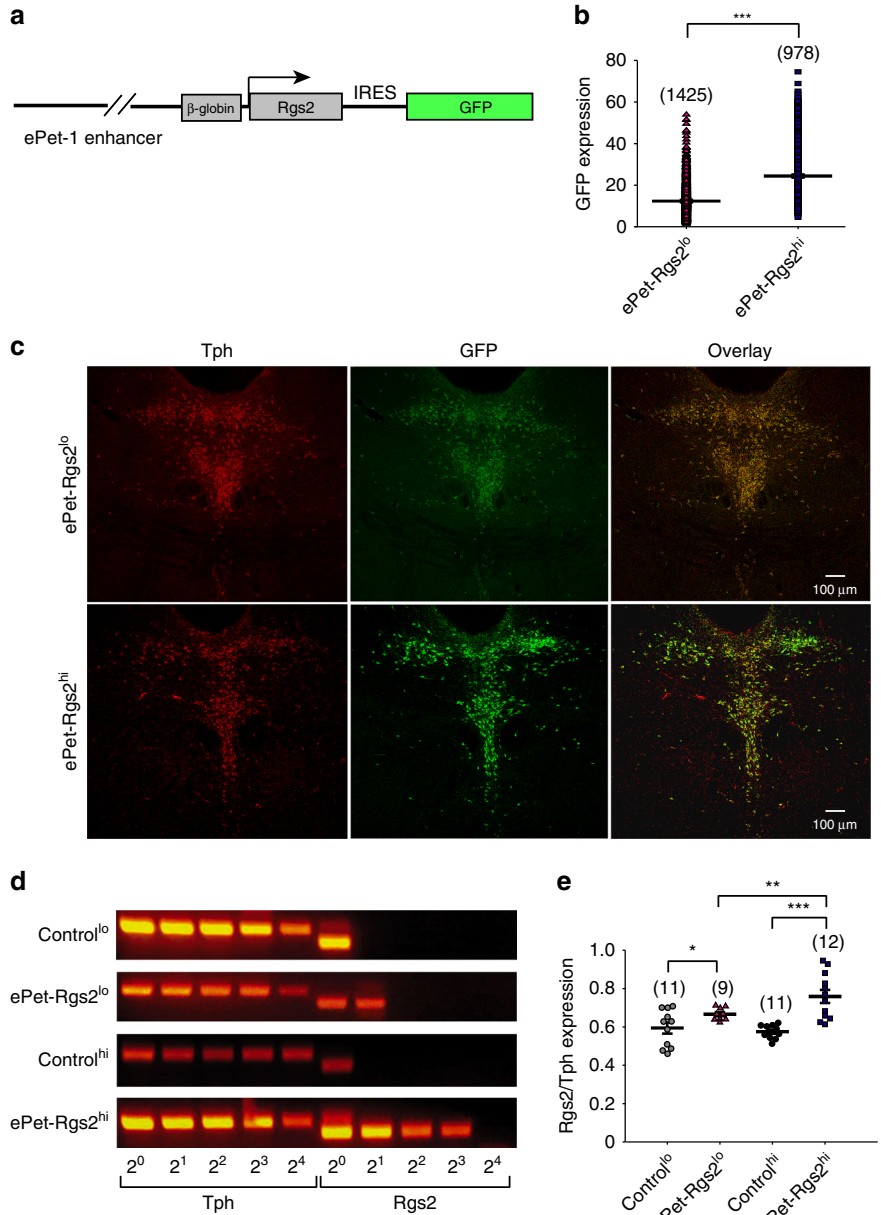

**Fig. 2** Creation of transgenic mouse lines overexpressing RGS2 and quantification of *Rgs2* mRNA levels specifically in serotonergic neurons. **a** Schematic of the construct used to create the BAC transgenic mouse lines, ePet-Rgs2lo and ePet-Rgs2hi. Rgs2-IRES-GFP was cloned 3′ of the 40 kb *ePet-1* enhancer sequence. **b** Relative GFP expression levels in serotonergic neurons from ePet-Rgs2lo and ePet-Rgs2hi mouse brains. **c** Expression of the serotonergic-specific marker *Tph*, Rgs2-IRES-GFP, and colocalization of *Tph* and Rgs2-IRES-GFP in the DRN from an ePet-Rgs2lo and ePet-Rgs2hi mouse. **d** Example gels from single-cell quantitative reverse transcriptase-PCR of *Rgs2* and *Tph* as internal control from YFP+, serotonergic neurons from littermate controllo, ePet-Rgs2lo, littermate controlhi, and ePet-Rgs2hi mouse brains. Serial dilutions from $2^0$ to $2^4$ were prepared from reversely transcribed cDNA. Single-cell qRT-PCRs revealed higher *Rgs2* mRNA expression in ePet-Rgs2hi compared to ePet-Rgs2lo mice. **e** Quantitative analysis of the relative *Rgs2* mRNA compared to *Tph* from serotonergic-positive neurons from control and ePet-Rgs2 transgenic mice indicate a 20.6% increase in *Rgs2* mRNA levels in ePet-Rgs2hi compared to ePet-Rgs2lo mice. The number of cells analyzed is reported within parentheses for each mouse line. Relative gene expression data reported as mean ± SEM. Statistical significance was evaluated with ANOVA (*$p < 0.05$, **$p < 0.01$, ***$p < 0.001$).

both ePet-Rgs2 transgenic mouse lines showed robust aggressive behavior compared to control littermates. From 54 trials with randomized opponents, 55% of the ePet-Rgs2lo and 70.4% of the ePet-Rgs2hi male mice forced their control littermates out of the tube (Fig. 3g, h, Supplementary Movie 3). Together, these results suggest that exogenous, serotonergic-specific expression of RGS2 in a dose-dependent manner leads to more aggressive but not anxious behavior in male mice.

**Rescue of RGS2 in 5HT neurons recovers male aggression in *Rgs2*−/− mice.** RGS2 deficiency in mice leads to an increase in anxiety and decline in male aggression[7]. To determine if the serotonergic system is involved in RGS2-dependent male aggression, we rescued RGS2 expression specifically in serotonergic neurons of *Rgs2*−/− mice by crossing them with the ePet-Rgs2 transgenic mouse lines. Both rescue lines showed increased number and duration of attacks in the resident intruder

**Table 1 Anxiety and depression tests for RGS2 overexpression mouse lines**

| Parameter | Control[lo] (11) Mean ± SEM | ePet-Rgs2[lo] (14) Mean ± SEM | *p* Value | Control[hi] (12) Mean ± SEM | ePet-Rgs2[hi] (11) Mean ± SEM | *p* Value |
|---|---|---|---|---|---|---|
| Open field | | | | | | |
| Duration in border (s) | 735.8 ± 24.4 | 718.5 ± 23.6 | 0.71 | 765.8 ± 33.9 | 727.9 ± 32. | 0.09 |
| Duration in center (s) | 164.2 ± 19.4 | 181.5 ± 10.4 | 0.41 | 134.2 ± 8.9 | 172.1 ± 13.0 | 0.02 |
| Frequency in border | 11.6 ± 3.4 | 6.5 ± 1.1 | 0.12 | 11.2 ± 1.9 | 18.4 ± 6.8 | 0.30 |
| Frequency in center | 15.5 ± 2.3 | 14.6 ± 1.4 | 0.73 | 15.7 ± 1.4 | 22.5 ± 3.1 | 0.05 |
| Elevated plus maze | | | | | | |
| Duration in closed arms (s) | 256.9 ± 8.9 | 257.0 ± 6.4 | 1.0 | 251.7 ± 10.3 | 247.8 ± 12.6 | 0.61 |
| Duration in open arms (s) | 15.8 ± 1.6 | 15.7 ± 1.9 | 1.0 | 17.5 ± 1.7 | 19.1 ± 1.3 | 0.70 |
| Duration in center (s) | 27.3 ± 4.5 | 27.3 ± 3.6 | 1.0 | 30.8 ± 3.7 | 33.1 ± 4.2 | 0.72 |
| Frequency in closed arms | 34.8 ± 4.3 | 33.5 ± 3.4 | 0.81 | 35.6 ± 3.6 | 36.9 ± 2.9 | 0.83 |
| Frequency in open arms | 12.1 ± 1.7 | 11.2 ± 1.2 | 0.63 | 14.3 ± 1.2 | 16.8 ± 2.1 | 0.58 |
| Frequency in center | 63.3 ± 6.4 | 60.3 ± 6.8 | 0.75 | 60.7 ± 5.2 | 62.8 ± 6.5 | 0.67 |
| Place preference | | | | | | |
| Duration in dark zone (s) | 149.9 ± 15.5 | 156.1 ± 9.2 | 0.73 | 149.9 ± 15.5 | 156.1 ± 9.2 | 0.72 |
| Duration in light zone (s) | 150.1 ± 15.5 | 143.9 ± 9.2 | 0.72 | 150.1 ± 15.5 | 143.9 ± 9.2 | 0.72 |
| Frequency in dark zone | 8 ± 0.9 | 8.2 ± 0.5 | 0.83 | 8.0 ± 0.9 | 8.2 ± 0.5 | 0.83 |
| Frequency in light zone | 8.4 ± 0.8 | 8.9 ± 0.5 | 0.63 | 8.4 ± 0.8 | 8.9 ± 0.5 | 0.63 |
| Forced swim test | | | | | | |
| Distance moved (cm) | 1818.3 ± 91.3 | 1749.9 ± 67.2 | 0.54 | 2031.1 ± 73.7 | 2004.6 ± 129.9 | 0.86 |
| Duration immobile (s) | 117.8 ± 15.6 | 120.4 ± 8.6 | 0.89 | 116.9 ± 7.3 | 131.6 ± 13.8 | 0.35 |
| Duration mobile (s) | 241.4 ± 15.6 | 238.9 ± 8.5 | 0.89 | 239.8 ± 7.3 | 224.2 + 13.4 | 0.31 |
| Duration highly mobile (s) | 0.71 ± 0.18 | 0.69 ± 0.24 | 1.0 | 3.3 ± 0.5 | 4.2 ± 0.7 | 0.30 |
| Mean velocity (cm s$^{-1}$) | 4.6 ± 0.53 | 4.5 ± 0.39 | 0.86 | 5.6 ± 0.2 | 5.6 ± 0.4 | 0.86 |
| Tail suspension | | | | | | |
| Distance moved (cm) | 1310.4 ± 128.5 | 1158.8 ± 96.4 | 0.38 | 2235.9 ± 199.7 | 2308.5 ± 178.7 | 0.79 |
| Duration immobile (s) | 288.3 ± 8.3 | 299.4 ± 9.6 | 0.41 | 247.4 ± 11.8 | 245.4 ± 9.1 | 0.89 |
| Duration mobile (s) | 71.7 ± 8.3 | 60.6 ± 9.6 | 0.40 | 105.8 ± 11.7 | 110.7 ± 8.7 | 0.74 |
| Duration highly mobile (s) | 0.018 ± 0.02 | 0.011 ± 0.01 | 0.69 | 6.8 ± 3.2 | 3.8 ± 1.1 | 0.41 |
| Mean velocity (cm s$^{-1}$) | 3.6 ± 0.7 | 2.8 ± 0.37 | 0.10 | 6.3 ± 0.55 | 6.4 ± 0.50 | 0.84 |

test compared to $Rgs2^{-/-}$ mice (Fig. 4a–d, Supplementary Movies 4 and 5). Moreover, Rgs2 rescue lines demonstrated a decrease in latency to first attack compared to $Rgs2^{-/-}$ mice (Fig. 4e). Surprisingly, $Rgs2^{-/-}$/ePet-Rgs2[lo] rescue mice showed an increase in the number of bites, which was not seen in the $Rgs2^{-/-}$/ePet-Rgs2[hi] mice (Fig. 4f). Additionally, both rescue lines, $Rgs2^{-/-}$/ePet-Rgs2[lo] (78%) and $Rgs2^{-/-}$/ePet-Rgs2[hi] (69%), won the majority of their battles in the tube dominance test against their $Rgs2^{-/-}$ littermate controls (Fig. 5a–b, Supplementary Movie 6). Surprisingly, the rescue mice from line $Rgs2^{-/-}$/ePet-Rgs2[lo] won 43% of their battles against their exogenous expression ePet-Rgs2[lo] littermates, and $Rgs2^{-/-}$/ePet-Rgs2[hi] mice won 60% of their battles against their exogenous expression ePet-Rgs2[hi] siblings in the tube dominance test (Fig. 5c, d, Supplementary Movie 7). Similar to the Rgs2 exogenous expression lines, no changes in anxiolytic or depressive behavior was observed in the rescue lines compared to $Rgs2^{-/-}$ mice (Table 2). Together, these results show that rescue of RGS2 in only serotonergic neurons of $Rgs2^{-/-}$ mice is sufficient to recover male aggression but not anxiety, indicating that RGS2 plays a major role in driving male aggression via the serotonergic system.

**RGS2 overexpression in 5HT neurons increases neural activity.** Several studies have shown that pharmacological or genetic reduction of 5HT or their receptors, in particular the 5HT$_{1A}$R and 5HT$_{1B}$R, can lead to aggressive behavior in mice and men[34]. Chronic or acute reduction of serotonergic activity by exogenous expression of 5HT$_{1A}$Rs or treatment with agonists in mice also led to an enhancement of aggression and suppression of 5HT neuron firing[35]. To investigate the physiological impact of RGS2 exogenous expression in serotonergic neurons, we performed in vivo extracellular recordings from DRN neurons in anesthetized mice. These neurons were selected for broad spikes, slow and regular firing and should contain between 60 and 80% of serotonergic neurons[36–41]. DRN neurons from exogenously expressing ePet-Rgs2[hi] mice showed a higher firing frequency in contrast to $Rgs2^{-/-}$ mice, which demonstrated a lower frequency and precision of firing compared to controls (Fig. 6c–d). Moreover, exogenous expression of Rgs2 in $Rgs2^{-/-}$ serotonergic neurons was able to recover DRN (serotonergic) neuron frequency and precision of firing to control levels (Fig. 6c–d).

We were also able to confirm increases in neural activity by *c-fos* expression in the DRN after 7 days of resident intruder assays. Not surprising, control and *Rgs2* overexpressing and rescue lines displayed a similar rise in DRN *c-fos* induction (as seen in firing frequencies) following resident intruder assays compared to no resident intruder encounters, but decreases in $Rgs2^{-/-}$ mice (Fig. 7a, b). Two important studies using optogenetic and pharmacogenetic control of the VMHvl identified it as an important locus for eliciting aggressive behaviors in mice[42,43]. Furthermore, removing inhibitory inputs into the VMHvl glutamatergic cells by lesioning the lateral septum GABAergic inputs increased aggression in mice[44]. To investigate if the neuronal activity is augmented in the VMHvl of *Rgs2*-overexpressing mice, we measured *c-fos* levels in the VMHvl before and after resident intruder tests. Indeed, *c-fos* induction in control and *Rgs2*-overexpressing mice were robust in the VMHvl (Fig. 7a, c). $Rgs2^{-/-}$ mice displayed a decrease in VMHvl *c-fos* induction following resident intruder assays compared to the *Rgs2*-overexpressing and control lines. Collectively, these data show that exogenous expression of Rgs2 in serotonergic neurons enhances their firing frequencies and maintains their precision of firing. Furthermore, exogenous expression of Rgs2 in serotonergic neurons increases the overall neuronal activity of the DRN and VMHvl following aggressive induced behavior in mice.

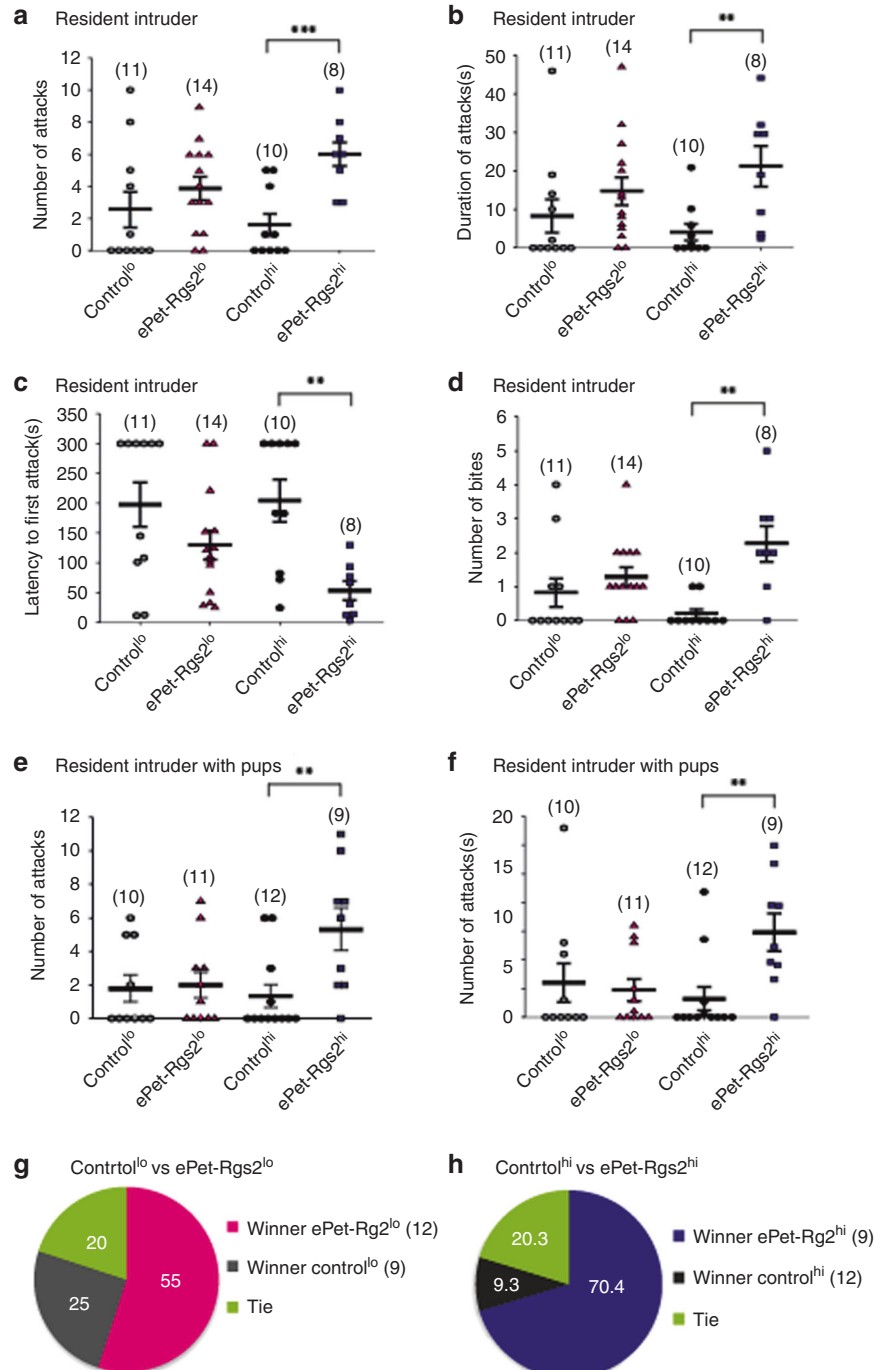

**Fig. 3** Exogenous expression of RGS2 in serotonergic neurons induces aggressive behavior in mice. Exogenous expression of RGS2 in serotonergic neurons in ePet-Rgs2hi mice induce aggressive behavior in the resident intruder test without (**a**–**d**) and with pups (**e**, **f**) compared to their littermate controls (controllo, grey circles; controlhi, black circles). In the resident intruder test without pups ePet-Rgs2hi (squares) mice but not ePet-Rgs2lo (triangles) showed an increase in the number of attacks (**a**), an increase in the duration of attacks (**b**), a decrease in latency to first attack (**c**), and an increase in the number of bites (**d**) compared to control and ePet-Rgs2lo mice. The ePet-Rgs2hi mice also displayed an increase number (**e**) and duration (**f**) of attacks in the resident intruder test with pups compared to control and ePet-Rgs2lo mice. The number of mice tested is indicated within parentheses. Data are reported as mean ± SEM. Statistical significance was evaluated with ANOVA (**$p < 0.01$, ***$p < 0.001$). **g** Aggression in the tube dominance test where mouse pairs are released at opposite ends of a plastic tube. Winners displaced or forced their opponent (loser) out of the tube. Aggression was measured in paired groups of males, littermate controllo (grey) vs. ePet-Rgs2lo (magenta) (**g**) and littermate controlhi (black) vs. ePet-Rgs2hi (dark blue) (**h**). Results are indicated as the percentage of displacement per group (winners). Aggressive behavior was pronounced in transgenic mice overexpressing RGS2 in serotonergic neurons. Six trials/mouse were performed with random opponents. The number of mice tested/group is indicated within parentheses

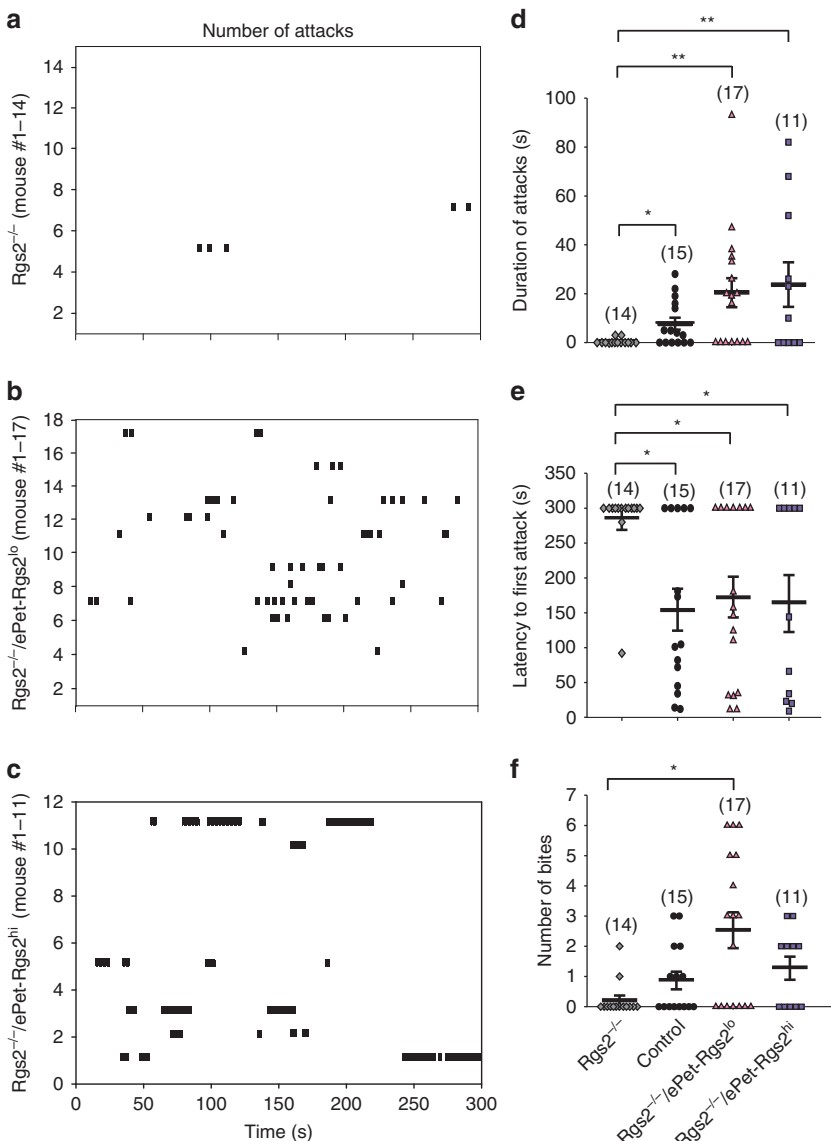

**Fig. 4** Rescue of RGS2 in serotonergic neurons recovers aggression in resident intruder test. Increased number and duration of attacks over a 10 min testing period from Rgs2-rescued mice, $Rgs2^{-/-}$/ePet-Rgs2$^{lo}$ (**b**) and $Rgs2^{-/-}$/ePet-Rgs2$^{hi}$ (**c**) compared to $Rgs2^{-/-}$ mice (**a**) on resident intruder day 10. The number of mice tested/group is indicated within parentheses. The average duration of attacks (**d**), latency to first attack (**e**), and number of bites (**f**) from $Rgs2^{-/-}$ mice (diamonds) compared to control (circles), Rgs2-rescued, $Rgs2^{-/-}$/ePet-Rgs2$^{lo}$ (triangles), and $Rgs2^{-/-}$/ePet-Rgs2$^{hi}$ (squares) mice are depicted as bar graphs. Rgs2-rescued mice showed augmented number and duration of attacks and reduced latency to first attack compared to their $Rgs2^{-/-}$ littermates. As expected, control mice demonstrated increased duration of attacks and reduced latency to first attack compared to their $Rgs2^{-/-}$ littermates. The number of mice tested/group is indicated within parentheses. Data are reported as mean ± SEM. Statistical significance was evaluated with ANOVA (*$p < 0.05$, **$p < 0.01$)

**RGS2 expression in 5HT neurons attenuates GPCR-mediated Ca$^{2+}$ spikes**. Serotonergic activity is in particular regulated by 5HT$_{1A}$ and 5HT$_{1B}$ autoreceptors coupling to the G$_{i/o}$ pathway as well as other modulatory transmitter systems such as the noradrenergic system coupling to G$_{i/o}$ and G$_{q/11}$ pathways[27,37,40,45,46]. 5HT$_{1A}$ are localized somatodendritically and reduce serotonergic activity by activation of G protein-coupled inward-rectifying potassium channels (GIRK). 5HT$_{1B}$ receptors are localized at the presynaptic terminal, inhibit presynaptic Ca$^{2+}$ channels, and therefore 5HT release onto 5HT neurons, thus reducing auto-inhibition and increasing serotonergic activity. α$_1$ARs coupling to the G$_{q/11}$ pathway increase serotonergic activity[26]. RGS2 has been shown to block G$_{q/11}$ and accelerate and attenuate G$_{i/o}$ signals in

a receptor-specific manner[47]. Therefore, we investigated if RGS2 affects GPCR modulation of serotonergic activity. We infected DRN neurons with the Ca$^{2+}$ indicator GCaMP6 and compared Ca$^{2+}$ spikes before and after application of 5HT$_{1A}$, 5HT$_{1B}$, and α$_1$AR receptor (α$_1$AR, G$_{q/11}$-coupled GPCR) agonists in DRN brain slices (Fig. 8). We observed that a much higher percentage of GCaMP6-infected neurons were silent in brain slices from $Rgs2^{-/-}$ mice in comparison to wild type or ePet-Rgs2$^{hi}$ (Fig. 8a–c). Application of the 5HT$_{1A}$ agonist 8-hydroxy-DPAT (8-OH-DPAT) decreased the rate of Ca$^{2+}$ spikes in control$^{hi}$ and ePet-Rgs2$^{hi}$, as well as $Rgs2^{-/-}$ mice (Fig. 8d). In contrast, the 5HT$_{1B}$ agonist increased the firing rate drastically for $Rgs2^{-/-}$, moderately for control$^{hi}$ mice, and had no effect on ePet-Rgs2$^{hi}$

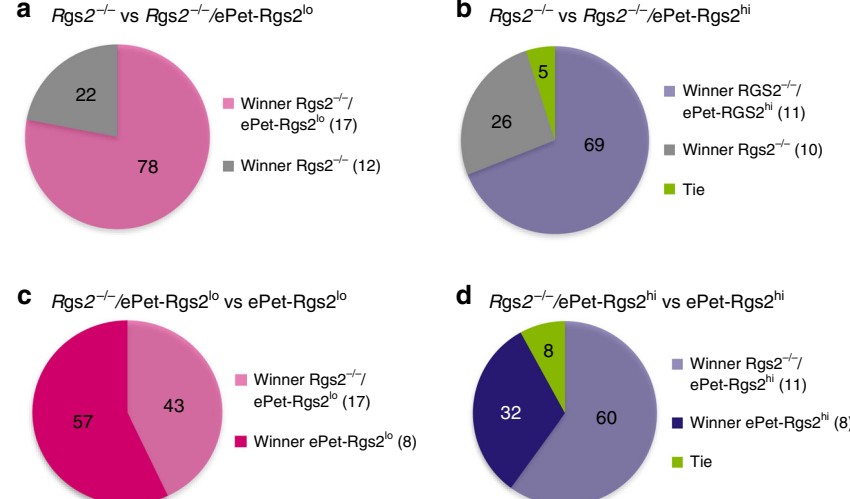

**Fig. 5** Rescue of RGS2 in serotonergic neurons recovers aggression in $Rgs2^{-/-}$ mice by the tube dominance test. Aggression was measured in combinations of paired groups from knockout ($Rgs2^{-/-}$), exogenous expression (ePet-Rgs2$^{lo}$, ePet-Rgs2$^{hi}$) and rescue lines ($Rgs2^{-/-}$/ePet-Rgs2$^{lo}$, $Rgs2^{-/-}$/ePet-Rgs2$^{hi}$). Mouse pairs were released at opposite ends of a plastic tube and the winners displaced or forced their opponent out of the tube. Paired groups of males included $Rgs2^{-/-}$ (grey) vs. $Rgs2^{-/-}$/ePet-Rgs2$^{lo}$ (light magenta) (**a**), $Rgs2^{-/-}$ (grey) vs. ePet-Rgs2$^{hi}$ (dark blue) (**b**), $Rgs2^{-/-}$/ePet-Rgs2$^{lo}$ (light magenta) vs. ePet-Rgs2$^{lo}$ (magenta) (**c**), and $Rgs2^{-/-}$/ePet-Rgs2$^{hi}$ (light blue) vs. ePet-Rgs2$^{hi}$ (dark blue) (**d**). Exogenous expression of RGS2 in serotonergic neurons rescued aggressive behavior in $Rgs2^{-/-}$/ePet-Rgs2$^{lo}$ mice and more dramatically in $Rgs2^{-/-}$/ePet-Rgs2$^{hi}$ mice. Results are indicated in the pie graphs as the percentage of displacement per group (winners). Four trials/mouse were performed with random opponents. The number of mice tested/group is indicated within parentheses

| Table 2 Anxiety and depression tests for RGS2 rescue mouse lines | | | | | | |
|---|---|---|---|---|---|---|
| **Parameter** | $Rgs2^{-/-}$ (14) **Mean ± SEM** | $Rgs2^{-/-}$/ePet-Rgs2$^{lo}$ (17) **Mean ± SEM** | $p$ **Value** | $Rgs2^{-/-}$ (10), **Mean ± SEM** | $Rgs2^{-/-}$/ePet-Rgs2$^{hi}$ (11) **Mean ± SEM** | $p$ **Value** |
| Open field | | | | | | |
| Duration in border (s) | 889.5 ± 4.6 | 888.3 ± 4.3 | 0.41 | 888.5 ± 3.6 | 887.2 ± 5.8 | 0.66 |
| Duration in center (s) | 10.5 ± 2.0 | 11.7 ± 2.2 | 0.71 | 11.5 ± 1.2 | 12.8 ± 2.2 | 0.62 |
| Frequency in border | 109.1 ± 11.0 | 90.6 ± 8.4 | 0.18 | 96.0 ± 9.9 | 94.9 ± 10.8 | 0.92 |
| Frequency in center | 18.4 ± 2.4 | 15.9 ± 1.8 | 0.39 | 14.1 ± 1.2 | 12.5 ± 1.2 | 0.38 |
| Elevated plus maze | | | | | | |
| Duration in closed arms (s) | 268.7 + 5.9 | 279.8 ± 4.6 | 0.14 | 240.8 ± 13.8 | 276.6 ± 11.7 | 0.33 |
| Duration in open arms (s) | 9.2 ± 3.0 | 5.1 ± 1.8 | 0.24 | 11.2 ± 1.6 | 6.8 ± 0.94 | 0.23 |
| Duration in center (s) | 21.9 ± 3.1 | 15.1 ± 3.6 | 0.18 | 37.4 ± 3.6 | 32.0 ± 2.8 | 0.55 |
| Frequency in closed arms | 18.3 ± 2.4 | 12.3 ± 1.3 | 0.03 | 8.8 ± 1.0 | 8.5 ± 1.1 | 0.84 |
| Frequency in open arms | 4.4 ± 1.3 | 1.5 ± 0.5 | 0.03 | 3.1 ± 0.73 | 3.5 ± 0.86 | 0.77 |
| Frequency in center | 20.7 ± 3.1 | 13.3 ± 1.7 | 0.04 | 22.3 ± 3.8 | 22.5 ± 4.1 | 1.0 |
| Place preference | | | | | | |
| Duration in dark zone (s) | 226.1 ± 9.1 | 222.6 ± 9.1 | .79 | 109.1 ± 17.5 | 81.4 ± 14.9 | 0.24 |
| Duration in light zone (s) | 73.9 ± 9.1 | 77.4 ± 9.1 | 0.80 | 190.9 ± 17.5 | 218.5 ± 14.9 | 0.24 |
| Frequency in dark zone | 6.8 ± 0.7 | 6.1 ± 0.6 | 0.50 | 6.9 ± 0.7 | 6.1 ± 1.0 | 0.51 |
| Frequency in light sone | 6.9 ± 0.8 | 6.2 ± 0.6 | 0.46 | 7.6 ± 0.8 | 6.8 ± .7 | 0.47 |
| Novelty suppressed feeding | | | | | | |
| Latency to eat (s) | 112.1 ± 18.3 | 68.7 ± 11.7 | 0.05 | 57.3 ± 16.8 | 111.5 ± 23.8 | 0.08 |
| Food consumption (g) | 0.98 ± 0.015 | 0.084 ± 0.0085 | 0.39 | 0.072 ± .02 | 0.03 ± 0.007 | 0.052 |
| Forced swim test | | | | | | |
| Distance moved (cm) | 760.3 ± 50.3 | 750.5 ± 39.6 | 0.89 | 1668.1 ± 109.0 | 1714.7 ± 78.0 | 0.73 |
| Duration immobile (s) | 70.8 ± 10.8 | 78.9 ± 8.1 | 0.54 | 134.4 ± 22.5 | 116.3 ± 14.7 | 0.50 |
| Duration mobile (s) | 283.2 ± 10.2 | 277.2 ± 7.5 | 0.63 | 224.1 ± 22.5 | 242.5 ± 14.8 | 0.50 |
| Duration highly mobile (s) | 6.3 ± 1.2 | 4.8 ± 0.9 | 0.38 | 1.3 ± 0.2 | 1 ± 0.3 | 0.42 |
| Mean velocity (cm s$^{-1}$) | 6.4 ± 0.4 | 6.3 ± 0.3 | 0.89 | 4.7 ± 0.3 | 4.8 ± 0.2 | 0.84 |
| Tail suspension | | | | | | |
| Distance moved (cm) | 1880.7 ± 148.8 | 1399.4 ± 90.2 | 0.08 | 869.3 ± 120.0 | 1149.5 ± 152.2 | 0.17 |
| Duration immobile (s) | 250.5 ± 9.6 | 280.8 ± 7.2 | 0.02 | 315.8 ± 7.3 | 303.4 ± 7.5 | 0.23 |
| Duration mobile (s) | 108.9 ± 9.3 | 79.1 ± 7.2 | 0.02 | 41.8 ± 6.8 | 54.8 ± 6.9 | 0.20 |
| Duration highly mobile (s) | 0.9 ± 0.6 | 0.4 ± 0.2 | 0.05 | 0.23 ± 0.1 | 1.6 ± 1.0 | 0.18 |
| Mean velocity (cm s$^{-1}$) | 5.2 ± 0.4 | 3.9 ± 0.2 | 0.08 | 2.4 ± 0.3 | 3.2 ± 0.4 | 0.16 |

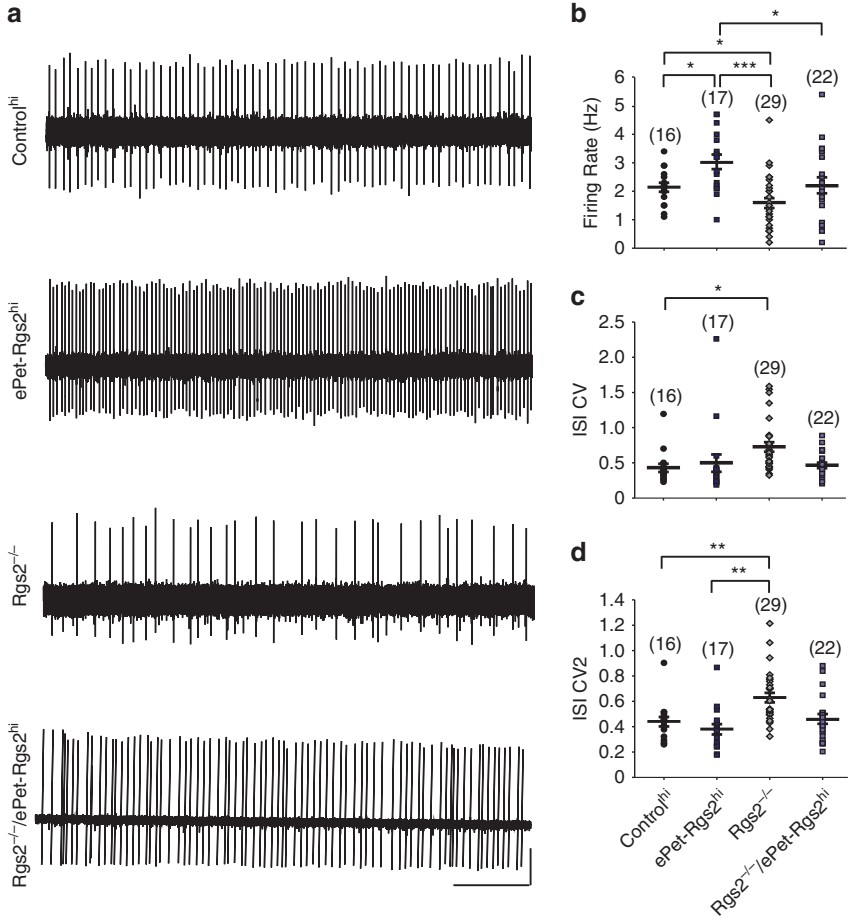

**Fig. 6** Enhanced frequency of serotonergic neuron firing in mice overexpressing RGS2 specifically in seronergic neurons. **a** Representative traces from control[hi], ePet-Rgs2[hi], $Rgs2^{-/-}$, and $Rgs2^{-/-}$/ePet-Rgs2[hi] serotonergic neuron firing patterns. Traces were measured from extracellular in vivo recordings of serotonergic neurons in anaesthetized mice (scale bar: voltage [a.u.] arbitrary units, 5 s). The mean firing rates (**b**), coefficient of variation of mean interspike interval (CV; **c**), and coefficient of variation of adjacent intervals (CV2; **d**) were analyzed from recorded cells. Serotonergic neurons from ePet-Rgs2[hi] (dark squares) mice exhibited a higher firing frequency compared to $Rgs2^{-/-}$ (grey diamonds) serotonergic neurons, which displayed slower firing rates and increased CVs. The number of cells analyzed are reported within parentheses for each mouse line. Mice ($\geq 3$) were tested per line. Data reported as mean ± SEM. Statistical significance was evaluated with ANOVA (*$p < 0.05$, **$p < 0.01$, ***$p < 0.001$)

(Fig. 8e). Application of the $\alpha_1$AR agonist also increased $Ca^{2+}$ spikes in $Rgs2^{-/-}$ and wild-type slices, but not in ePet-Rgs2[hi] (Fig. 8f). Thus, exogenous expression of RGS2 selectively modulates intrinsic GPCR signals in serotonergic neurons.

## Discussion

In the present study, we demonstrate that RGS2, a member of the small RGS family, drives male aggression through the serotonergic system. Exogenous expression of RGS2 exclusively in serotonergic neurons elevated the duration and number of attacks, decreased the latency of attacks, and increased the displacement of male mice in the tube dominance test. Moreover, we were able to rescue the docile phenotype in $Rgs2^{-/-}$ mice by overexpressing RGS2 only in serotonergic neurons. Aside from ePet-Rgs2[hi] mice spending more time in the center of the open field test, we did not observe any changes in anxious or depressed behavior in the $Rgs2$-overexpressing and rescue mouse lines compared to their respective control littermates. Our data identify a critical neuronal population, that is, serotonergic neurons in the DRN, where RGS2 modulates male aggression but not anxiety.

This result is in particular surprising, since RGS2 has been implicated to be involved in anxiety- and depression-related disorders in mice and men. Mouse models knocking out $Rgs2$

show increased anxiety and decreased aggression, and, more recently, heightened neophobia and fear learning[7,21,48]. Furthermore, extensive human genetic association studies implicate $Rgs2$ to anxiety, panic, and post-traumatic stress disorders[10–16]. Previous investigations suggest that individuals with depressed $Rgs2$ levels may be potentially at risk for anxiety- or depressive-like related disorders, which can manifest aggressive behavior. A patient study from suicide victims observed increased levels of RGS2 in prefrontal cortex and amygdala regions, which may suggest that anxiety- and depression-related disorders in humans could be mediated through other areas of the brain[49]. Moreover, linkage studies in humans with RGS6, 7, 8, 5, 19, and 12 implicate their involvement in anxiety, depression, bipolar disorders, and/or schizophrenia. Since only $Rgs2$, 13, 14, and G$\alpha$-interacting protein ($Gaip$) mRNAs were found in the DRN rat brain, other RGS proteins may be mediating their depressive and anxious behaviors outside the DRN, such as via the nucleus accumbens, amygdala, prefrontal cortex, or hippocampus[49,50]. Besides a plethora of data linking changes in 5HT levels to anxiety phenotypes, our data suggest that RGS2 modulates anxiety outside the serotonergic system.

The use of germ-line knockout mice for our rescue experiments with a transgenic line may present problems due to the

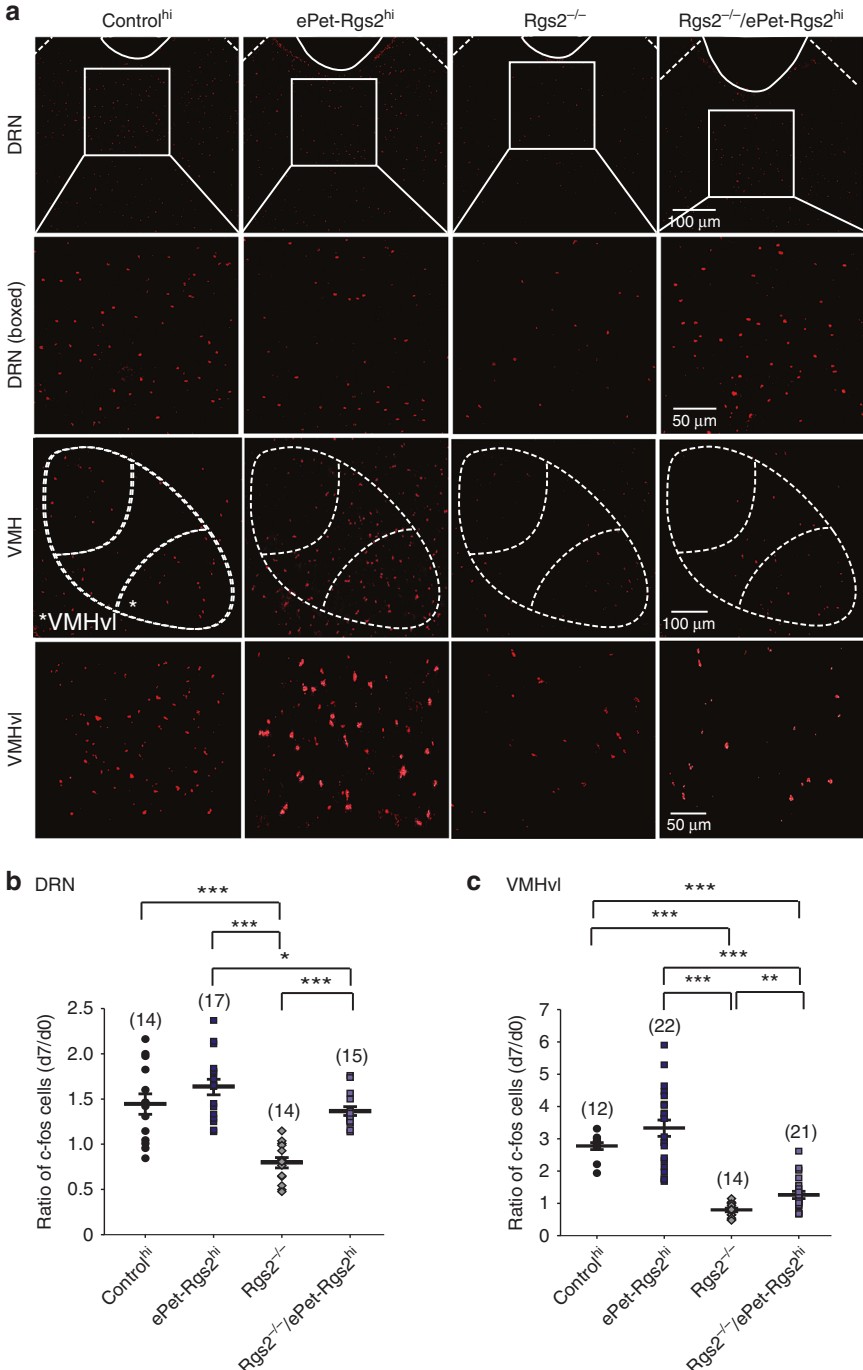

**Fig. 7 Exogenous expression of RGS2 in serotonergic neurons induces *c-fos* expression in the dorsal raphe nucleus (DRN) and ventrolateral part of the ventromedial nucleus hypothalamus (VMHvl).** (**a**) Representative images of the DRN, higher magnification of boxed DRN area, VMH, and a higher magnification of the VMHvl area from control[hi], ePet-Rgs2[hi], $Rgs2^{-/-}$, and $Rgs2^{-/-}$/ePet-Rgs2[hi] mice following 7 days of resident intruder tests. Asterisk represents the analyzed VMHvl region. Increased relative number of cells expressing *c-fos* after 7 days of resident intruder testing compared to day 0 (naive) in the DRN (**b**) and VMHvl (**c**) from control[hi] (black circles), ePet-Rgs2[hi] (dark squares), $Rgs2^{-/-}$/ePet-Rgs2[hi] (light squares), and $Rgs2^{-/-}$ (grey diamonds) mice. The number of slices analyzed is reported within parentheses for each mouse line. Mice ($\geq 3$) were tested per line. Data were reported as the average total number of *c-fos*-positive cells/slice within parentheses for each genotype, mean ± SEM. Statistical significance was evaluated with ANOVA (*$p < 0.05$, **$p < 0.01$, ***$p < 0.001$)

different original backgrounds of the mice. For example, the $Rgs2^{-/-}$ mice originate from a 129 strain and the transgenic *Rgs2* overexpression lines from a mixed background. Although they were both backcrossed for more than 17 generations in the C57/Bl6 background, genes from the original knockout, 129 strain in the flanking region and genetic background, or position effects

from the transgenic line may contribute to the phenotype and physiology of the knockout and rescue mice[51,52]. Another caveat of behavior studies is the use of littermate controls and nulls, which may alter aggressive behavior due to the postnatal influences from their siblings[52]. Hence, the interpretation and translation of the data must be made with caution due to potential

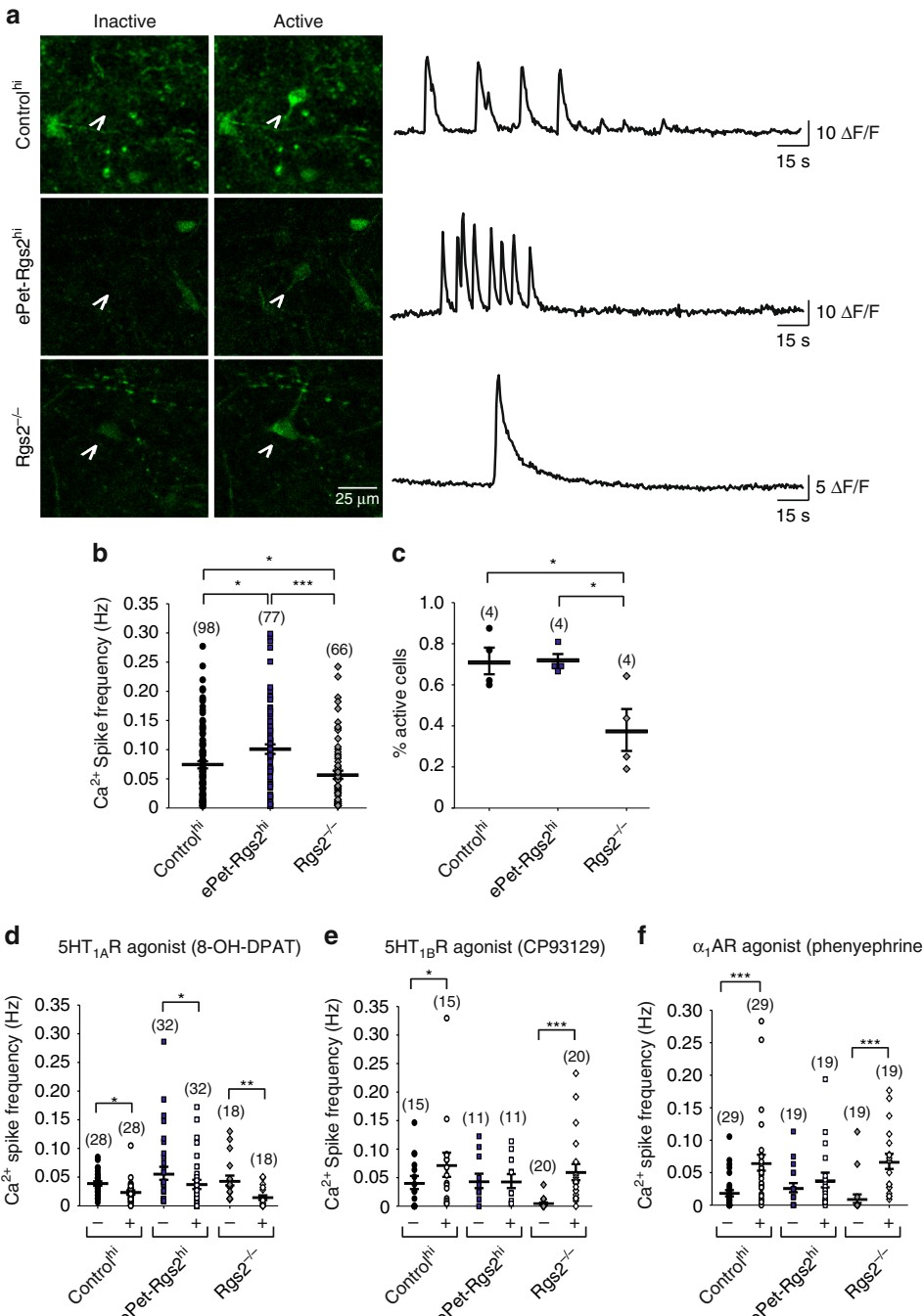

**Fig. 8** Exogenous expression of RGS2 in serotonergic neurons accelerates GPCR-mediated calcium signals in the DRN. **a** Representative calcium images and time-dependent changes in calcium fluorescence from a control[hi], ePet-Rgs2[hi], and *Rgs2−/−* cell in the DRN during inactive and active states. *Rgs2−/−* cells showed minimal changes in calcium levels compared to increased calcium signals in control[hi] and ePet-Rgs2[hi] cells in the DRN. **b** Calcium spike frequency and **c** percentage of active cells in the DRN from control[hi] (circles), ePet-Rgs2[hi] (squares), and *Rgs2−/−* (diamonds) slices. (Note that neurons with a $Ca^{2+}$ half spike width >5 s were excluded from calculating the mean calcium spike frequency in **b**; Supplementary Fig. 1A–C.) Calcium spike frequency of cells in the DRN before (closed, −) and after (open, +) application of **d** 5HT$_{1A}$R (8-OH-DPAT), **e** 5HT$_{1B}$R (CP93129), and **f** α$_1$ adrenergic (phenyephrine) receptor agonist from control[hi] (circles), ePet-Rgs2[hi] (squares), and *Rgs2−/−* (diamonds) slices. Number of cells is reported within parentheses for graphs **b** and **d**–**f**. Number of independent experiments is reported within parentheses for graph **c**. Data are reported as mean ± SEM. Statistical significance was evaluated with ANOVA (*$p < 0.05$, **$p < 0.01$, ***$p < 0.001$)

developmental adaption and physiological compensation secondary to the loss of the key gene during development[53].

We observed in our study that RGS2 determines the spontaneous activity and precision of serotonergic neuron firing in vivo. High expression of RGS2 increases the precision and frequency of serotonergic firing, while serotonergic neurons with low levels/no

expression of RGS2 reveal a lower, more irregular firing frequency. We also observed that the level of expression of RGS2 determines the amount of 5HT$_{1B}$, but not 5HT$_{1A}$, modulation of $Ca^{2+}$ spikes. 5HT$_{1B}$ receptors are localized at the presynaptic terminal of serotonergic neurons. 5HT$_{1B}$ receptors activate the G$_{i/o}$ pathway and inhibit presynaptic $Ca^{2+}$ channels leading to a

reduction of 5HT release. Axon collaterals of serotonergic projections back propagate onto 5HT neurons and downregulate the spontaneous activity of 5HT neurons. Therefore, $5HT_{1B}$ receptor agonists increased DRN cell firing in wild-type mice, but not in $5HT_{1B}$-knockout mice[54,55]. In addition, expression of RGS2 at presynaptic terminals of serotonergic neurons predict an increase in the 5HT release, since RGS2 has been shown to increase synaptic transmitter release probability by downregulating the $G_{i/o}$-mediated presynaptic $Ca^{2+}$ channel inhibition at hippocampal autapses[5]. The downregulation of the $G_{i/o}$-mediated presynaptic $Ca^{2+}$ channel inhibition is also reflected in the absence of the $5HT_{1B}$ agonist effects in ePet-Rgs2[hi] mice since RGS proteins inhibit G protein signaling and/or the acceleration of signal termination[56] (see below).

Based on our previous study, we hypothesize that 5HT release in serotonergic neurons in the presence of RGS2 is increased and involves the RGS2-mediated attenuation of $5HT_{1B}$ receptor responses. However, this has to be demonstrated in in vivo recordings from behaving animals and has to be compared to studies of $5HT_{1B}$ autoreceptor knockout mice, which reveal an anti-anxiety and anti-depressant phenotype rather than an aggressive phenotype, which is observed in $5HT_{1B}$ forebrain-knockout mice[57,58].

RGS2 not only accelerates the $G_{i/o}$ pathway but also act as a GTPase-activating protein on the $G_{q/11}$ pathway[47]. In fact, expression of RGS2 in *Xenopus* oocytes completely suppresses the $G_{q/11}$-induced activation of the $Ca^{2+}$-activated $Cl^{-}$ conductance[59]. We observed a similar effect for the $\alpha_1AR$ receptor modulation of $Ca^{2+}$ spike activity in serotonergic neurons, which is abolished by a high expression of RGS2. RGS2 has been shown to specifically interact with $\alpha_1AR$ receptors via sphinophilin. This RG2/spinophilin interaction drastically attenuate/block adrenaline-induced $\alpha_1AR$ receptor signaling in a concentration-dependent manner, which is not observed in cells from $Rgs2^{-/-}$ mice[60].

RGS proteins act as GTPase-activating proteins, which accelerate the hydrolysis of bound GTP on the G protein α-subunit. Physiologically this leads either to the inhibition of G protein signaling or the acceleration of signal termination[56]. Importantly, modulation of the GPCR signal can be GPCR/RGS specific, leading to the control of a subset of GPCR signal within a neuronal/cellular population in RGS concentration-dependent manner[61]. Since Rgs2 is an IEG, which is dynamically upregulated during activity and targets specific GPCRs and signaling pathways, one can hypothesize that the upregulation of RGS2 in 5HT neurons, for example, during stress, will shift the serotonergic modulation in the brain to higher aggression.

For example, the locus coerulus (LC) and the raphe nuclei reveal extensive inhibitory interconnectivity. $5HT_2$ receptors on the prolactin-releasing hormone nucleus drive the main inhibitory input to the LC. In return, the LC acts via inhibitory $\alpha_2$ and excitatory $\alpha_1$ adrenoceptors on the DRN[62]. Since RGS2 seems to dampen adrenergic signaling (Fig. 8) and increase 5HT release, one could postulate an increased 5HT tone within the aggression circuit[5].

There is an abundance of reports in humans and mice that led to the hypothesis that low 5HT levels as a trait-like phenomenon are associated with aggressive behavior[24,34,63,64]. For example, patients with high level of aggressive and violent behavior have low cerebrospinal fluid levels of 5HT metabolites[20]. Deletion of the majority of 5HT neurons in the mouse brain leads to a reduction of brain 5HT levels and an increase in aggression[25].

In contrast, during the performance and expression of aggressive behavior, 5HT levels have been shown to be increased[63]. For example, activation of $GABA_A$ or $5HT_{1A}$ receptors in the DRN, which leads to the inhibition of 5HT

neuronal activity, decreases aggressive behaviors, and high-aggressive vs. low-aggressive rats reveal an increased number of *c-fos*-positive 5HT neurons in the DRN and medial raphe nuclei categorized by their performance in the residence intruder test[65]. Rats and mice exposed to aggressive and agonistic encounters also reveal an increase in *c-fos*-positive 5HT neurons or DRN neurons[66,67]. In addition, acquisition of dominance behavior in female hamsters has been associated with activation of DRN 5HT neurons and $5HT_{1A}$ agonists injected into the hypothalamus or systemically administered fluoxetine increases aggression (note that male hamsters reveal the opposite behaviors)[68]. Social interaction with an unfamiliar partner and area also results in an increase in extracellular 5HT levels in the ventral hippocampus[69].

Species-specific aggression as tested with the resident intruder and tube dominance test involves a stress (neuroendocrine) response and can be divided into different behavioral states, that is, predisposition, social engagement, aggressive interaction, and establishment of social rank[70]. Aggressive interaction induces an immediate stress response accompanied with high 5HT levels in both dominant and subordinate animals. It has been suggested that dominant animals have immediate and stronger endocrine (i.e. 5HT) responses during social engagement, the time before aggressive interactions. Therefore, our results may suggest that RGS2 is increasing 5HT release in particular during initiation of aggression. Since RGS2 is speeding up GPCR signals by accelerating the GTPase activity of, in particular, $G_{i/o}$ proteins and attenuating a specific subset of GPCR signals, 5HT neurons should respond much faster and with higher amplitudes to changes in the neuroendocrine response selected for increased aggressive behavior.

We observed in our study that *c-fos* staining in the DRN is increased after aggressive behavior in ePet-Rgs2[hi]-expressing mice, and decreased in $Rgs2^{-/-}$ mice. Surprisingly, there was only a tendency of increased *c-fos* expression in ePet-Rgs2[hi]-expressing mice compared to control[hi]-expressing mice, despite persistently augmented aggressive behavior. Since the aggressive behavior of ePet-Rgs2[hi] mice peaked after 7–8 days of resident intruder encounters, perhaps *c-fos* expression levels adapted or peaked earlier in both lines as described previously in a similar study in rats[71]. In this study, they reported changes in the *c-fos* expression patterns in specific regions of the brain after 10 vs. 1 encounter to a resident intruder. Certain brain regions remained unaltered, while others increased in *c-fos* expression. Interestingly, the *c-fos* activity in the DRN continued to increase following the 10th encounter. *C-fos* is an IEG and activated by increased neuronal activity. Thus, the results suggest that the DRN neuronal circuit is activated by aggression. Increased *c-fos* expression in serotonergic neurons has also been observed in highly aggressive vs. low aggressive rats after the resident intruder test[65]. We observed and increased *c-fos* staining in serotonergic and non-serotonergic neurons in the DRN.

Upregulation of 5HT neuronal activity during aggression may involve glutamatergic and GABAergic input into the DRN and recruitment of local feedforward and feedback circuits. For example, increased glutamate release causes a phasic increase of 5HT release from DRN 5HT neurons and is observed during escalated aggression. The process may involve activation of $GABA_B$ receptors localized on non-serotonergic neurons in the DRN, which leads to an increase in glutamate release[72].

The ventromedial prefronal cortex (PFC) sends direct glutamatergic input into the DRN, which drives *c-fos* expression and synaptic transmitter release, particularly in GABAergic neurons[73]. However, other studies suggest that synaptic input into the DRN from the PFC is bilaterally organized and innervates both GABAergic and 5HT neurons[74]. In addition, other brain areas have been identified, which directly innervate the DRG and

reveal increased *c-fos* activity during aggressive behavior, such as the PFC, the ventromedial and lateral hypothalamus, the amygdala, and the locus coeruleus[75,76]. These areas send glutamatergic projections (PFC and LhB) and GABAergic and glutamatergic projections (lateral hypothalamus, preoptic area, substantia nigra) to GABAergic and 5HT neurons in a brain region-specific manner with defined spatial organization within the DRN[74,77].

## Methods

**Real-time qPCR**. To compare the relative mRNA levels of *Rgs2* and *Rgs4* in hippocampal and serotonergic neurons, total RNA was isolated from 21 days in culture hippocampal or embryonic 12.5 (E12.5) serotonergic neurons with RNeasy Mini Kit (Qiagen Inc.) and purified with on-column DNase digestion using RNase-Free DNase Set (Qiagen Inc.). Continental cultures of hippocampal neurons were prepared according to a modified version of published procedures from mouse pups (P0–3)[78,79]. Dissociated serotonergic neurons were previously enriched with FACS (of live cells) from ePet-YFP+ mice where serotonergic neurons are expressing YFP[30].

For real-time PCR, 1 mg of RNA was used for reverse transcription with Advantage Real Time for PCR Kit (BD Biosciences) to generate 100 μl complementary DNA (cDNA). Three microliters of the final real-time product was used for real-time PCR for *Rgs2*, *Rgs4*, *ePet-1* (serotonergic specificity), *Tph2* (serotonergic specificity), and *β-actin* (loading control). 18S RNA was used in a 1:100–300 dilution as internal control and normalized to the 18S RNA found in neurons from wild-type mice. Real-time PCR quantification was performed on iCycler Iq Detection System (Bio-Rad) with CYBR Green Assay (Bio-Rad), and cDNA fragments of *Rgs2*, *Rgs4*, *ePet-1*, *Tph2*, *β-actin*, and *18S* RNA were amplified with primer pairs: *Rgs2* forward/reverse: TGATTGCCCAAAATATCCAA/GGGCTCCGTGGTGATCTG; *Rgs4* forward/reverse: AGAAATGGGCTGAATCGTTG/CCTCTCTGGTGCAAGAGTCC; *ePET-1* forward/reverse: GCAGCGGGCAGATCCAGTTG/TGAGCTTGAACTCGCCGTGG; *Tph2* forward/reverse: TTTAAGGACAATGTCTATCG/CTGGGAATGGGCTGACCATA; *β-actin* forward/reverse: CCATCTTGGCCTCACTGTCC/AGCTCAGTAACAGTCCGCCT; *18S* forward/reverse: AAACGGCTACCACATCCAAG/CCTCCAATGGATCCTCGTTA. The PCR reactions used a modified two-step profile with an initial denaturation step for 3 min at 95 °C, followed by 40 cycles of 15 s denaturation at 95 °C and 25 s polymerase reaction at 57 °C. The experiments were performed with three independent neuronal cultures in duplicates ($n = 6$). Specificity of real-time PCR products was documented with gel electrophoresis and resulted in a single product with desired length. The melt-curve analysis showed that each primer pair had a single product-specific melting temperature. All primer pairs have at least 95% of PCR efficiency as reported from the slopes of the standard curves generated iQ software (Bio-Rad, version 3.1). Relative gene expression data was analyzed with $2-\Delta\Delta CT$ method[80].

**Mice**. Adult male and female mice ($Rgs2^{-/-}$, ePet-Rgs2$^{lo}$, ePet-Rgs2$^{hi}$, $Rgs2^{-/-}$/ePet-Rgs2$^{lo}$, control$^{lo}$, $Rgs2^{-/-}$/ePet-Rgs2$^{hi}$, control$^{hi}$) were housed on a 12 h light/dark cycle with food and water ad libitum. The present study was carried out in accordance with the European Communities Council Directive of 2010 (2010/63/EU) for care of laboratory animals and approved by a local ethics committee (Bezirksamt Arnsberg) and the animal care committee of North Rhine-Westphalia, Germany, based at the LANUV (Landesamt für Umweltschutz, Naturschutz und Verbraucherschutz, Nordrhein-Westfalen, D-45659 Recklinghausen, Germany). The study was supervised by the animal welfare commission of the Ruhr-University Bochum. All efforts were made to minimize the number of mice used for this study.

**Trangenic mice**. The $Rgs2^{-/-}$ mice were a gift from Josef M. Penninger at the Institute of Molecular Biotechnology in Vienna, Austria and backcrossed into C57/Bl6 mice[7]. For stable integration of Rgs2 constructs into the mouse genome, Rgs2-IRES-GFP was subcloned 3′ to the *β-globin* minimal promoter and 40 kb serotonergic-specific *ePet-1* enhancer in the modified pBACe3.6 vector as previously described[29]. The pBAC-ePet-Rgs2-IRES-GFP constructs were microinjected into pronuclei by the Case Western Reserve University Transgenic Core Facility (Cleveland, OH, USA). Transgene expression was detected by PCR analysis (forward/reverse: TCTGAACAGGAGCCCATCCC/TTATGCCCAGCCCATCGAAT) and GFP expression in founders. We received nine founder lines, which were positive for the ePet-Rgs2-IRES-GFP BAC constructs. These founder lines were backcrossed for >17 generations into C57/Bl6 mice and evaluated for RGS2/GFP expression in 5HT neurons. Two high-expressing ePet-Rgs2-IRES-GFP founder lines were bred to hemizygosity.

**Histology**. Fluorescence microscopy was performed as previously described[81]. Adult male mice (six ePet-Rgs2$^{lo}$ mice, five ePet-Rgs2$^{hi}$ mice) were perfused with 4% paraformaldehyde in phosphate-buffered saline (PBS), pH 7.4, and dissected brains post fixed in the same solution for 1 h. Brains were cryoprotected in 30%

sucrose (wt vol$^{-1}$), 1× PBS overnight at 4 °C. Thirty micrometers of coronal cryostat sections were collected, blocked in 5% fetal bovine serum (FBS) (fetal bovine serum) in PBST (PBS with 0.1% Triton X-100) for 1 h at room temperature and incubated with a primary antibody against GFP (1:1000 dilution, Frontier Institute) and conjugated with goat-anti-rabbit-A488 secondary antibody (1:1000; Molecular Probes). Images of brain slices were analyzed with a Leica TCS SP5II confocal microscope using a ×20/0.3 NA objective. The intensity of the GFP fluorescence in serotonergic neurons was quantified using NIH Image J.

**Single-cell qRT-PCR**. Coronal slices (250 μm thick) through the DRN were prepared from male mice by using a modified Leica VT1000S vibratome. Slices were incubated at 30 °C for 25 min and then maintained at room temperature. During recordings, slices were superfused with artificial cerebrospinal fluid (ACSF; 124 mM NaCl, 3 mM KCl, 1.23 mM NaH$_2$PO$_4$, 26 mM NaHCO$_3$, 10 mM dextrose, 2.5 mM CaCl$_2$, and 1.2 mM MgSO$_4$) equilibrated with 95% O$_2$, 5% CO$_2$ at 30 °C (flow rate, 1–2 ml min$^{-1}$)[82]. Coronal slices were prepared from different *Rgs2* transgenic mouse lines (≥3 male mice/genotype) expressing YFP under the control of *Pet-1* enhancer to allow for better visualization of serotonergic neurons by YFP. To isolate mRNA from YFP+ serotonergic neurons (>3 neurons/mouse), coronal slices were maintained and equilibrated with 95% O$_2$, 5% CO$_2$ at 33 °C in external solution (12.5 mM NaCl, 0.25 mM KCl, 0.125 mM NaH$_2$PO$_4$, 2.6 mM NaHCO$_3$, 2 mM CaCl$_2$, 1 mM MgSO$_4$, 2 mM D(+)glucose, pH 7.4). Pulled glass micropipettes (2–4 MΩ) filled with internal solution (125 mM potassium gluconate, 10 mM HEPES, 4 mM NaCl, 0.2 mM EGTA, 2 mM MgCl$_2$, 4 mM ATP, 0.4 mM GTP, 10 mM phosphocreatine, 10 mM KOH, pH 7.3) were used to patch individual neurons and remove cytoplasm-containing mRNA. cDNA was transcribed with SuperScript III First Strand Synthesis System with oligo (dT) primers according to the manufacturer's instructions (Invitrogen, Life Technologies). Serial dilutions from $2^0$ to $2^4$ were prepared from reversely transcribed cDNA. *Rgs2* and *Tph* (internal loading control) were amplified using Red Load Taq Master (Jena Bioscience) from cDNAs by quantitative reverse transcription-PCR (qRT-PCR). Duplicate qRT-PCRs were performed for each cell. Relative RNA expression levels were calculated as a ratio of the *Rgs2* and *Tph* band intensities with NIH Image J. The background intensity of each gel was also factored into the calculation to eliminate exposure differences between gels. The following oligonucleotide primers were used for qRT-PCR: *Rgs2* forward/reverse: GGCAGAAGCATTTGATGAAC/TGAAGCAGCCACTTGTAGCC; *Tph* forward/reverse: TGTGGCCATGGGCTATAAAT/TGTAGAGGGGGTCGGAGC.

**Behavior tests for aggression**. *Resident intruder test*: Resident male mice were single housed for at least 14 days before testing and at least 5 days without cage cleaning to facilitate territorial behavior. Resident mouse cages were not cleaned for the entire testing session. Six- to eight-week-old resident intruder mice were housed in rat cages in groups of 15 male mice per cage for at least 10 days before testing to allow acclimation to the dark/light cycle. For 10 consecutive days, one male intruder was introduced to a resident intruder cage for 5 min during their dark cycle. The time to first attack, number of attacks, duration of attacks, number of threats, number of tail rattles, and number of bites were video tracked with the Ethovision XT 8.5 software (Noldus) and later analyzed in a blinded fashion.

*Resident intruder test with pups*: Resident male mice were housed with one female until pups were born. Resident's female and pups (postnatal 7–10 days old) were separated from the resident with a clear plastic barrier before testing. Resident mouse cages were not cleaned for the 10 days of testing. Six- to eight-week-old resident intruder mice were housed in rat cages in groups of 15 male mice per cage for at least 10 days before testing to allow acclimation to the dark/light cycle. For 10 consecutive days, one male intruder was introduced to a resident intruder cage for 5 min during their dark cycle. The female and pups were present during the trial and separated from the males by an olfactory permeable partition. The time to first attack, number of attacks, duration of attacks, number of threats, number of tail rattles, and number of bites were video tracked with the Ethovision XT 8.5 software (Noldus) and later analyzed in a blinded fashion.

*Tube dominance test*: Aggression was tested by using displacement from a plastic tube (30 cm long and 3.0 cm diameter). Pairs of male mice were released into either end of the tube. The first mouse to exit was the "loser." A maximum session of 10 min was performed and 6 trials with randomized opponents. In the event that both mice remained in the tube for the entire 10 min or both mice exited the tube at the same time, this was considered a "tie." The plastic tube was cleaned between subjects with 70% ethanol. Number of winners per group is represented as a percentage of the total number trials.

**Behavior tests for anxiety**. *Open field test*: The open field arena consisted of an acrylic chamber (30 × 30 × 30 cm$^3$; Noldus), subdivided into a center (14.5 × 14.5 cm$^2$) and border region, which was brightly illuminated by several 75 W incandescent bulbs mounted above the arena. Mice were placed into the center of the open field and the following parameters were video tracked for 15 min with the Ethovision XT 8.5 software (Noldus): time spent in the center, time spent in the border, total distance traveled, and border-to-center transitions. The apparatus was cleaned between subjects with 70% ethanol. For each mouse data were averaged from three trials.

*Elevated plus maze test*: A modified version of the elevated plus maze was designed according to Pellow et al[83]. A maze consisting of two, $33 \times 6 \text{ cm}^2$ open arms, two $33 \times 6 \times 16.5 \text{ cm}^2$ closed arms, and a $6 \times 6 \text{ cm}^2$ open center was elevated 43 cm above the floor. Mice were acclimated to the behavior lab 60 min before testing. During the 5 min testing period, the following parameters were video tracked and analyzed with Ethovision XT 8.5 software (Noldus): number of entries into the closed arms, open arms and center, time spent in the closed arms, open arms and center, number of head-dips, number of urine puddles, and number of fecal pellets. The maze apparatus was cleaned between subjects with 70% ethanol. For each mouse data were averaged from three trials.

*Place preference test*: The place preference test, also known as the light/dark mouse exploration test was created according to the modified specifications as Crawley and Goodwin[84]. Briefly an open field arena ($30 \times 30 \times 30 \text{ cm}^3$) was divided into two arenas, an open, light arena and a dark closed arena consisting of a black infrared see through plexiglass box with an opening ($30 \times 15 \times 30 \text{ cm}^3$). The light arena was brightly illuminated by 75 W incandescent bulbs mounted above the arena. Mice were placed into the right corner light arena and the following parameters were video tracked with an infrared camera for 5 min with the Ethovision XT 8.5 software (Noldus): time spent in the light and dark arena and the number of transitions between the light and dark arenas. The arenas were cleaned between subjects with 70% ethanol. For each mouse data were averaged from three trials.

*Novelty suppressed feeding test*: Mice were deprived of food for 24 h before testing with water available ad libitum. A familiar food pellet (weighing ~2 g) was placed in the middle of a new aversive environment (arena: $30 \times 30 \times 30 \text{ cm}^3$), brightly illuminated with two 75 W incandescent bulbs. Mice were placed into a plastic tube in the right corner of the arena, which was removed at the start of the test. The task ended when the mice first fed, defined as biting the food pellet with use of the forepaws. The latency to start feeding served as a measurement of anxious behavior. Subjects were recorded by a video camera for further analysis. Mice were allowed to consume food an additional 5 min in the arena before returning to home cages. Food consumption was measured for potential feeding differences among test groups.

**Behavior tests for depression**. *Force swim test*: Mice were placed in a 5 L beaker with a 17 cm diameter and water (23–25 °C) to a depth of 20 cm[85]. Mice were placed in the water undisturbed for duration of 6 min. Each mouse underwent three trials. The time spent swimming (mobile vs. highly mobile) vs. the time spent floating (immobile) was video tracked with the Ethovision XT 8.5 software (Noldus). The number of fecal pellets per session were also recorded and disposed of after each trial.

*Tail suspension test*: Mice were resuspended by their tails using adhesive tape for 6 min[86]. Each mouse underwent three trials. Cumulative mobility (mobile vs. highly mobile) vs. immobility time was video tracked with the Ethovision XT 8.5 software (Noldus).

All transgenic ePet-Rgs2 mouse lines used for behavior studies were hemizygotes. Four- to six-month-old male mice were used for behavior experiments. All mice were single housed in the behavior lab for the duration of the testing period. For the aggression tests, male mice were single housed for at least 4 weeks before testing. Cleaning of the cages was executed at least 2 days prior to testing and not during the testing period to facilitate territorial behavior. Anxiety tests were performed during the light cycle, whereas the aggression tests were performed during the dark cycle of the mice. Mice acclimated 14 days to the dark/light cycle and behavior lab before testing. Data acquisition was performed using a video recording system and Ethovision tracking software. All statistical analyses were calculated by means of one-way analysis of variance (ANOVA). Significance for comparisons: $*p < 0.05$, $**p < 0.01$; $***p < 0.001$. Results are presented as mean ± SEM from all trials.

**Extracellular in vivo recordings of serotonergic neurons**. Recordings have been previously performed and described[87–89]. Briefly, adult male mice (≥3 mice/genotype) were anesthetized with 1.5–2% isoflurane. A sagittal incision along the mid-line was performed to expose the cranium and a window 4.0 mm caudal from bregma to access the dorsal raphe region. A multi-electrode device ("Eckhorn microdrive," Thomas Recording, Giessen) was used to monitor single neurons within the DRN. Single-cell activity was sampled with high temporal resolution (32 kHz) and analyzed offline for spike detection. Alternatively, signals of multiple electrodes were routed through an on-line spike sorter (Plexon, Dallas, TX) and time markers for each detected action potential were stored. Recordings were performed in naive, anesthetized animals in the dorsal raphe. The position of the dorsal raphe was determined according to the *xyz* coordination of mouse brain (anterior-posterior (AP): −4.1; medial-lateral (ML): 0.0; dorso-ventral (DV: −1.7 to 2.5 mm).

In the offline data analysis, single unit action potentials were detected with custom-made software implemented in Matlab. Possible serotonergic neurons were identified using the classic criteria, broad spikes and slow (around 2 Hz), regular firing, established previously[36,37]. According to Beck et al.[38] and Kirby et al.[38,39] error rates using AP duration as a criteria to distinguish serotonergic from non-serotonergic neurons in the DRN are around 20–40%.

Baseline spike trains were analyzed for mean firing rate and spike width. To quantify the spike train regularity, coefficients of variation (CV) of interspike intervals (ISIs) were calculated. Additionally, the CV for adjacent intervals, CV2 ([2 $(\Delta t_{i+1} - \Delta t_i)] (\Delta t_{i+1} + \Delta t_i)^{-1})$, of ISIs were calculated[88]. An average of CV2 over $n$ estimates the intrinsic variability of a spike train, nearly independent of slow variations in average rate. All statistical analyses were calculated by means of one-way ANOVA. Significance for comparisons: $*p < 0.05$, $**p < 0.01$; $***p < 0.001$. Results are presented as mean ± SEM. Exponential fit of activity after stimulation onset and offset was made with IgorPro (WaveMetrics, Portland, OR, USA).

**C-fos induction studies**. Rescue mouse line $Rgs2^{-/-}/ePet-Rgs2^{hi}$, littermate control[hi], $Rgs2^{-/-}$, and ePet-Rgs2[hi] mice were introduced to an intruder for 0 (naive) or 7 days. Ninety minutes after exposure to the intruder, mice were anesthetized and perfused with 4% paraformaldehyde in PBS, pH 7.4, and dissected brains were post fixed in the same solution for 1 h. Brains were cryo-protected in 30% sucrose (wt/vol), 1× PBS overnight at 4 °C. Forty-five micrometers of coronal sections were collected, blocked in 4% FBS in PBST for 1 h at room temperature and incubated with primary antibodies, rabbit anti-*c-fos* (1:1000 dilution, Santa Cruz), and secondary antibodies, goat-anti-rabbit Dylight 649 (1:1000; Molecular Probes), for 2.5 h at room temperature or overnight at 4 °C. Brain slices were mounted with Roti-Mount FlourCare DAPI (Roth) to identify the nucleus. Images of brain slices were analyzed with a Leica TCS SP5II confocal microscope using a ×20/0.3 NA objective. The number of slices analyzed is reported in the bars for each mouse line. Mice ≥3 were tested per line. Data were reported as the average total number of c-fos-positive cells/slice for each genotype, mean ± SEM. Statistical significance was evaluated with one-way ANOVA ($*p < 0.05$, $**p < 0.01$, $***p < 0.001$).

**Intracranial virus injections**. GCaMP viruses were injected in anaesthetized adult mouse brains in the dorsal raphe[90,91]. All mouse lines were injected with pAAV.Syn.GCaMP6f.WPRE.SV40 (Penn Vector Core, Philadelphia, PA), except for ePet-cre mice, which were injected with pAAV.CAG.Flex.GCaMP6f.WPRE.SV40 (Penn Vector Core, Philadelphia, PA) for 5HT-specific neuron expression. The position of the dorsal raphe was determined according to the *xyz* coordination of mouse brain (AP: −4.1; ML: 0.0; DV: −1.7 to 2.5 mm). Mice were deeply anesthetized with 1.5–2.0% isoflurane and placed into a stereotactic frame (Narishige, Japan). The skin was opened with a sagittal incision along the mid-line. A small craniotomy was performed and 0.5–1 µl of virus was applied in 100 µm steps using pressure injection in 2 min intervals. A customized glass pipette attached to a 5 ml syringe was used for virus delivery. At the end of injection the skin was sutured (Surgicryl Monofilament, Belgium). After the surgery, animals received subcutaneous injection of carprofen (2 mg kg$^{-1}$) for analgesia. Animals were placed individually into their home cages to recover and allow for virus expression for 7–14 days.

**GCamP6 imaging of DRN neurons in brain slices**. Coronal slices (250 µm thick) through the median raphe nuclei were prepared from male mice by using a modified Leica VT1000S vibratome as described above. Coronal slices were placed in a recording chamber with oxygenated (95% O$_2$, 5% CO$_2$) ACSF. All experiments were performed at 30 °C with a perfusion flow rate of 1–2 ml min$^{-1}$ [82]. GCaMP fluorescent neurons were visualized and imaged with a Leica TCS SP5 confocal laser scanning microscope interfaced to a personal computer, running Leica Application Suite Advanced Fluorescence software (LAS AF 2.6) with ×20 objective lens. GCaMP was excited with a 488 nm Argon laser. Emitted light was detected by a PMT after passing through a bandpass filter (505–605 nm). The confocal aperture was wide open during Ca$^{2+}$ imaging experiments to collect maximum emitted fluorescence. Imaging was performed at 1400 Hz, 371 ms frame$^{-1}$, and 512 × 512 frames. Drugs were dissolved in ACSF. To determine whether GPCR agonists were able to exhibit evoked Ca$^{2+}$ elevations, agonists were added directly to ACSF. CP93129 (5HT$_{1B}$ agonist, 50 nM), 8-OH-DPAT (5HT$_{1A}$ agonist, 10 mM), and phenyephrine ($\alpha_1$-AR agonist, 10 µM) were obtained from Tocris. Data were collected from at least four different experiments, using tissue derived from at least four mice for each condition. Ca$^{2+}$ spikes were detected and analyzed using dedicated software written in Matlab. The fluorescence trace was calculated by measuring the mean fluorescence intensity of the region of interest per frame. Regions of interest were marked before calculating $\Delta F/F$. $\Delta F/F = (F - F_0)/F_0$, where $F$ is the fluorescence and $F_0$ is the baseline fluorescence. The measured fluorescent intensity was corrected for a baseline value defined as the lowest (<10%) value of the fluorescence measured during the experiment. Ca$^{2+}$ spikes were analyzed offline and were defined as spikes with a clear onset and offset. The half-width of the spikes were calculated at the median of the spike amplitude.

**Statistics and reproducibility**. All statistical analyses were calculated with SigmaPlot and Microsoft Excel software unless otherwise stated. Data were initially analyzed for normality by the Shapiro–Wilk test ($p \geq 0.05$), and then tested for equal variance with the Equal Variance Test ($p \geq 0.05$). If data sets passed both tests, a *t* test for comparison of two groups or one-way ANOVA (post hoc *t* test) for comparison of more than two groups was used. Significance for comparisons:

*$p \le 0.05$, **$p \le 0.01$; ***$p \le 0.001$. Cell counts from specific areas of $n \ge 3$ mice are presented as mean ± SEM. For counting, slices were imaged for c-fos-positive cells, followed by manual cell count using NIH Image J. For calcium imaging, spikes were analyzed using NIH Image J (Fiji). In vivo recordings were analyzed by the Matlab software. The $n$ for every experiment is reported in the figures.

**Reporting summary**. Further information on research design is available in the Nature Research Reporting Summary linked to this article.

## Data availability
The authors declare that all the data supporting the findings of this study are available in the manuscript, figures, and supplementary information files. Source data can be found in Supplementary Data 1. All materials and other data supporting this study are readily available from the corresponding author upon reasonable request.

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

## Acknowledgements

We thank Stefan Dobers, Stephanie Krämer, Margareta Möllmann, Gina Weber, and Manuela Schmidt for excellent technical assistance and Christi Wylie and Mian Xie for help with some experiments. This work was supported by DFG MA 5806/2-1 (to M.D.M.), MA 5806/1-2 (to M.D.M.) and He2471/23-1 (to S.H.), He2471/21-1 (to S.H.), He2471/19-1 (to S.H.), Priority Program (SPP1926; to S.H.), Project number 122679504-SFB874 project B10 (to S.H.) and Project number 316803389-SFB1280 project A07 (to S.H.), and NIH/NIMH MH081127-A1 (to S.H.).

## Author contributions

The questions and experimental concept were developed by M.D.M. and S.H. Experiments were designed and conducted by M.D.M., E.S.D. and J.H. for creation of mice, by A.G., T.M., M.D.M. and E.S.D. for qRT-PCR, by A.G., T.M., M.D.M. and K.K. for histology and *c-fos* induction, by M.D.M., K.B. and C.J. for behavior, by P.W. and T.M. for electrophysiology and by S.H. and P.W. for calcium imaging. Data analyses and interpretations of data were conducted by all authors. S.H. and M.D.M. wrote the paper with contributions from all authors.

## Competing interests

All authors declare no competing interests.
