## [Peer Review File · Communications Biology]

Reviewers' comments:

Reviewer #1 (Remarks to the Author):

In the current submission, the authors describe the role of RGS2, a GAP protein, in controlling aggressive behavior, through an effect on the central serotonergic system. While prior reports have shown how RGS2 mutations are linked with increased anxiety behavior in humans, mouse mutations suggest this protein affects anxiety and aggression levels. Through overexpression and rescue experiments, the authors are able to link serotonergic neuron dysfunction and RGS2 signaling with the regulation of aggression. Replication of the reported results should be achievable using the described mouse genetic lines by other researchers. The reviewer believes that the manuscript contributes significantly to our understanding of how perturbations in serotonin neuron function can lead to changes in aggressive behavior. The reviewer suggests that only minor changes be made to manuscript in its current form.

Requested Changes:

1. In Figure 7, it appears that there is no significant cFos expression difference between controls and the RGS2HI animals following social defeat stress exposure. It would be important to address this point in the discussion.
2. In Figure 8, it is unclear whether an increase in calcium signal was seen when comparing control and RGS2HI neurons. If there is no difference, this point should be addressed in the discussion. Finally, a statistical comparison of silent vs active neuron numbers between the three compared treatments would be helpful.

Michael Scott
Associate Professor of Pharmacology
University of Virginia

Reviewer #2 (Remarks to the Author):

This study examines the effect of over-expression of RGS2 in 5-HT neurons of wild-type and RGS2 knockout mice. RGS2 inhibits G protein regulation of firing, and leads to activation of 5-HT neurons and an aggression phenotype in male mice, with no effect detected in anxiety/depression assays. The studies are consistent with a role for RGS2 in 5-HT neurons to promote aggressive behavior.

The paper is carefully done and presents convincing results on the role of RGS2 over-expression in 5-HT neurons. Strengths of the paper include the use of Lo vs. Hi RGS2 overexpressing mice, and rescue studies in the RGS2^{-/-} background. However, they do not directly address the importance of endogenous RGS2 in 5-HT neurons with a specific knockout or knockdown of RGS2 in 5-HT neurons.

Specific comments

1. Introduction: this could be shortened for focus on RGS2/5-HT, and less on MAO-A.
2. Methods: Describe post-tests used for multiple comparisons. Full statistical comparison

all groups should be presented in figures.

3. Methods: For electrophysiological studies, it would be important to verify that neurons recorded were 5-HT by GFP or TPH staining. When were the recording performed: before or after the resident intruder assays, or on naïve mice? Where was recording done: median or dorsal raphe?

4. Methods: How was GCaMP6 administered? Was it specifically expressed in 5-HT neurons? Provide evidence of what proportion of infected cells were 5-HT neurons.

5. Results: Description of the transgenic lines could be moved to Methods.

6. Fig. 1A: There are over 20 RGS proteins, yet the authors focus only on RGS2 and 4; it would be informative to address the relative levels of other RGS proteins in 5-HT neurons.

7. Fig. 2D: define the X-axis: is this the dilution factor?

8. Fig. 4D-F: Were duration, latency or number of bites different between control and RGS2-lo or hi? Comment on why only the attack duration (and # of bites for LO) seems to show a difference from control under the RGS2 background: it suggests that RGS2 may act at other sites or developmentally to partly drive the phenotype.

9. Fig. 6B: Was there a correlation of firing rate with ISI? What is r^2 and significance? Is this correlation meaningful?

10. Fig. 6C-E: Were the Firing rate, ISI different between RGS2hi vs. RGS2hi rescue and any other groups? In particular, RGS^{-/-} rescue should be compared with RGS^{-/-} to show a rescue.

11. Fig. 7: Was cFos increased in 5-HT, GABA or both cell type in DRN? Co-staining with TPH or 5-HT could address this. As a control for VMHvl, quantify cFos⁺ cells in another adjacent part of the hypothalamus.

12. Fig. 8: How were Ca spikes detected? How were cells labeled with GCaMP6, what virus, how, which cells? Were the inactive cells 5-HT⁺, non-5-HT or both cells?

13. Given the large number of RGS proteins, specific inhibition of RGS2 may not have a pronounced effect on 5-HT neurons due to compensation by other RGS proteins. Although changes were seen in the RGS2 knockout, these changes could involve altered activity of inputs to 5-HT neurons, rather than in 5-HT neurons. The authors could perhaps address the role of endogenous RGS2 using a viral knockdown strategy. This would be important given their main conclusion that targeting RGS2 in 5-HT neurons could be a therapeutic target.

14. Discussion: In contrast to the model suggested for 5-HT_{1B} autoreceptor signaling to reduce 5-HT release in the raphe, enhance raphe firing and induce aggression, more recent work indicates that knockout 5-HT_{1B} receptors in the forebrain mediate aggressivity, while knockout of 5-HT_{1B} autoreceptors have an anti-anxiety or anti-depressant effect (Nautiyal et al., 2015; Nautiyal et al., 2016). Thus the effect of RGS2 on aggressivity may not be driven by inhibition of 5-HT_{1B} autoreceptors.

References:

Nautiyal KM, Tritschler L, Ahmari SE, David DJ, Gardier AM, Hen R (2016) A Lack of Serotonin 1B Autoreceptors Results in Decreased Anxiety and Depression-Related Behaviors. *Neuropsychopharmacology* 41:2941-2950.

Nautiyal KM, Tanaka KF, Barr MM, Tritschler L, Le Dantec Y, David DJ, Gardier AM, Blanco C, Hen R, Ahmari SE (2015) Distinct Circuits Underlie the Effects of 5-HT_{1B} Receptors on Aggression and Impulsivity. *Neuron* 86:813-826.

Reviewer #3 (Remarks to the Author):

The current MS of Mark et al., presents the results of a series of multidisciplinary experiments aimed to test the hypothesis that the regulator of G protein signaling 2 protein (RGS2) in DRN serotonergic neurons is causally involved in regulating anxious and aggressive behavior in mice. By employing state-of-the-art genetic manipulation, behavioral monitoring and neurophysiological recording techniques, they convincingly demonstrate that overexpressing of RGS2 exclusively in DRN serotonergic neurons enhanced aggressive but not anxious behavior. Moreover, they were also able to rescue the docile phenotype of RGS2 knock-out mice by overexpressing RGS2 in serotonergic neurons. In addition, they nicely showed that the enhanced aggressive behavior in DRN RGS2 overexpressing mice is associated with enhanced neuronal activation (as assessed by the surrogate IEG marker c-fos) in dorsal raphe nuclei as well in the ventrolateral part of the VMH (shown previously as neuronal hotspot driving aggression), increased firing rate of serotonergic neurons and reduction of the modulatory actions of Gi/0 and Gq/11 – coupled 5-HT_{1B} autoreceptor and alpha1 adrenergic receptors in serotonergic neurons. Clearly, given the long-term link between the CNS serotonergic system and aggression/anxiety in both humans and animals, the objectives and rationale of this study is valid and comprehensibly described in the introduction. The aim and hypothesis of the study is lucid and to the point. As already mentioned, the employed methodology is contemporary and state-of-the-art using validated techniques that are meticulously conducted. The obtained results are interesting, novel and clearly presented in both text and graphs/tables. The discussion of and conclusions drawn from the results are similarly relevant and valid. In Toto, a very nice study with that is of great interest to the field. I have only a couple of points that may be addressed in the discussion.

1) How do the authors explain the lack of desensitisation of the 8-OHDPAT-induced decrease in calcium spike frequency in the DRN? The somatodendritic 5-HT_{1A} autoreceptors are coupled to the Gi/0 signaling pathway, just like the terminal 5-HT_{1B} autoreceptors. Given the observed increased spiking frequency of 5-HT neurons in the RGS2 overexpressor mice, one would expect this 5-HT_{1A} autoreceptor that importantly controls 5-HT neuronal activity to be desensitized.

2) The discussion section on the modulation of serotonergic activity and aggression states a lot of findings from other studies but no references are being given. These should be cited.

3) In the section on the neuronal activity in the DRN it is stated that the majority of increased c-fos staining was found in non-serotonergic neurons. However, given the presented results in figure 7 this does not seem to be the case as no double- staining with Tph or another specific serotonin cell marker has been performed. Hence, this claim should be removed or in case double-staining has been performed the results should be included. This aspect is particularly important as it may contribute to the somewhat dogmatic view that serotonergic neuronal activity becomes suppressed during the execution of aggressive behaviors. There is substantial (but poorly cited) evidence in the literature that the actual display of aggressive behavior is associated with robust but perhaps short-lasting increased 5-HT neuronal activity measured via different techniques (Van der Vegt et al., 2003; Veening et al., 2005; Haller et al., 2005; 2006; Ferris et al.,

2008; Terranova et al., 2016; Cadogan et al., 1994; Takahashi et al., 2011; 2015; Nakazato 2013; Agoy et al., 2013).

Reviewer #1 (Remarks to the Author):

In the current submission, the authors describe the role of RGS2, a GAP protein, in controlling aggressive behavior, through an effect on the central serotonergic system. While prior reports have shown how RGS2 mutations are linked with increased anxiety behavior in humans, mouse mutations suggest this protein affects anxiety and aggression levels. Through overexpression and rescue experiments, the authors are able to link serotonergic neuron dysfunction and RGS2 signaling with the regulation of aggression. Replication of the reported results should be achievable using the described mouse genetic lines by other researchers. The reviewer believes that the manuscript contributes significantly to our understanding of how perturbations in serotonin neuron function can lead to changes in aggressive behavior. The reviewer suggests that only minor changes be made to manuscript in its current form.

Requested Changes:

1. In Figure 7, it appears that there is no significant cFos expression difference between controls and the RGS2HI animals following social defeat stress exposure. it would be important to address this point in the discussion.

We added to the Discussion on p24 under the heading, "Species-specific aggression increases neuronal activity in the DRN.": ...Surprisingly, there was only a tendency of increased c-fos expression in ePet-Rgs2hi compared to controlhi expressing mice despite persistently augmented aggressive behavior. Since the aggressive behavior of ePet-Rgs2hi mice peaked after 7-8 days of resident intruder encounters, perhaps c-fos expression levels adapted or peaked earlier in both lines as described previously in a similar study in rats (Martinez, 1998). In this study they reported changes in the c-fos expression patterns in specific regions of the brain after 10 versus 1 encounter to a resident intruder. Certain brain regions remained unaltered while others increased in c-fos expression. Interestingly the c-fos activity in the DRN continued to increase following the 10th encounter.

2. In Figure 8, it is unclear whether an increase in calcium signal was seen when comparing control and RGS2HI neurons. If there is no difference, this point should be addressed in the discussion. Finally, a statistical comparison of silent vs active neuron numbers between the three compared treatments would be helpful.

In Fig 8B the calcium spike frequency is not significantly different between the control and RGS2HI neurons, but showed a trend for being significant ($p < 0.065$). We therefore analyzed more cells and included these data in the new Fig 8B on p44. The results reveal significant differences between ePet-Rgs2hi, Rgs2^{-/-} and controlhi. We also reanalyzed the percentage of active cells in the 4 independent experiments and report as mean \pm SEM in the new Fig 8C. The results reveal that for controlhi and ePet-Rgs2hi approximately 70% of the cells were active, whereas only 40% of the cells were active in the Rgs2^{-/-} mice. The differences in the amount of active cells is significant between controlhi and Rgs2^{-/-}, ePet-Rgs2hi and Rgs2^{-/-}, but not for controlhi and ePet-Rgs2hi. (see also reviewer #2, point 4)

Michael Scott

Associate Professor of Pharmacology
University of Virginia

Reviewer #2 (Remarks to the Author):

This study examines the effect of over-expression of RGS2 in 5-HT neurons of wild-type and RGS2 knockout mice. RGS2 inhibits G protein regulation of firing, and leads to activation of 5-HT neurons and an aggression phenotype in male mice, with no effect detected in anxiety/depression assays. The studies are consistent with a role for RGS2 in 5-HT neurons to promote aggressive behavior.

The paper is carefully done and presents convincing results on the role of RGS2 over-expression in 5-HT neurons. Strengths of the paper include the use of Lo vs. Hi RGS2 overexpressing mice, and rescue studies in the RGS2^{-/-} background. However, they do not directly address the importance of endogenous RGS2 in 5-HT neurons with a specific knockout or knockdown of RGS2 in 5-HT neurons.

Specific comments

1. Introduction: this could be shortened for focus on RGS2/5-HT, and less on MAO-A.

We agree with the reviewer and removed on p 4, first paragraph in the Introduction: For example the enzyme monoamine oxidase A (MAOA) catalyzes the oxidative deamination of 5HT at serotonergic presynaptic terminals to regulate its release and inactivation. Polymorphic variants in the MAOA gene, which lead to lower expression is associated with aggressive and violent behavior in humans (Brunner et al., 1993; Dorfman et al., 2014; Ficks and Waldman, 2014; Hunter, 2010). In agreement with the polymorphic studies in humans, knockout mice for MAOA also show increase reactive aggression (Scott et al., 2008).

2. Methods: Describe post-tests used for multiple comparisons. Full statistical comparison of all groups should be presented in figures.

We added to the Methods section under Statistics on p13:

Statistics

All statistical analyses were calculated with SigmaPlot software unless otherwise stated. Data were initially analyzed for normality by the Shapiro-Wilk test ($p \geq 0.05$), then tested for equal variance with the Equal Variance Test ($p \geq 0.05$). If data sets passed both tests, a t-test for comparison of two groups or One.Way ANOVA (post-hoc t-test) for comparison of more than two groups was used. Significance for comparisons: * $p \leq 0.05$, ** $p \leq 0.01$; *** $p \leq 0.001$. Cell counts from specific areas of $n \geq 3$ mice are presented as mean \pm SEM. For counting, slices were imaged for cfos⁺ cells followed by manual cell count using ImageJ.

Additionally in the Figure legends we state either "Data reported as mean \pm SEM. Statistical significance was evaluated with ANOVA (* $p < 0.05$, ** $p < 0.01$)." or "Statistical significance was evaluated with ANOVA (* $p < 0.05$, ** $p < 0.01$, *** $p < 0.001$)."

3. Methods: For electrophysiological studies, it would be important to verify that neurons recorded were 5-HT by GFP or TPH staining. When were the recordings performed: before or after the resident intruder assays, or on naïve mice? Where was recording done: median or dorsal raphe?

We agree with the reviewer that it would be nice to know the exact neuron type, from which we recorded from. However, we think that it is not feasible for the *in vivo* recordings we performed, because retrograde labeling using the Eckhorn matrix is very difficult to achieve and would result in a tremendous increase in sacrificed animals. Therefore we relied on classic criteria to select for serotonergic neurons as established previously, i.e. broad spikes and slow (around 2Hz) and regular firing (Aghajanian et al., 1978; Calizo et al., 2011; Mlinar et al., 2016; Vandermaelen and Aghajanian, 1983). According to Beck et al. 2004 and Kirby et al. 2003 error rates using AP duration as a criteria to distinguish serotonergic from nonserotonergic neurons in the DRN are around 20-40%. The error rate resembles the amount of 38% of non-serotonergic neurons identified in the DRN. Since we cannot distinguish between serotonergic and non-serotonergic neurons *in vivo*, we now say:

Methods

p11 last paragraph in Extracellular *in vivo* recordings: "A multi electrode device ("Eckhorn microdrive", Thomas Recording, Giessen) was used to monitor single neurons within the DRN.

p11 last paragraph in Extracellular *in vivo* recordings: Recordings were performed in naïve, anaesthetized animals in the dorsal raphe. The position of the dorsal raphe was determined according to the xyz coordination of mouse brain (AP: Between -4.24-4.9, ML: \pm 0.25, DV: -1800-3000 μ M).

p12 Possible serotonergic neurons were identified using the classic criteria...

p12 Please note, that according to Beck et al. 2004 and Kirby et al. 2003 (Beck et al., 2004; Kirby et al., 2003) error rates using AP duration as a criteria to distinguish serotonergic from non-serotonergic neurons in the DRN are around 20-40%.

In addition, we now give information about the mice we used and the position of the recordings. We now say on p11-12:

Recordings were performed in naïve, anaesthetized animals in the dorsal raphe. The position of the dorsal raphe was determined according to the xyz coordination of mouse brain (AP: Between -4.24-4.9, ML: \pm 0.25, DV: -1800-3000 μ M). (see also our own publication (Masseck et al., 2014; Spoida et al., 2014).

Results

Exogenous expression of RGS2 in serotonergic neurons leads to increases in neuronal firing and c-fos induction in the DRN and VMHvl.

p 18 To investigate the physiological impact of RGS2 exogenous expression in serotonergic neurons, we performed *in vivo* extracellular recordings from DRN neurons in anesthetized mice. These neurons were selected for broad spikes and slow and regular firing and should contain between 60-80% of serotonergic neurons (Aghajanian et al., 1978; Beck et al., 2004; Calizo et al., 2011; Kirby et al., 2003; Mlinar et al., 2016; Vandermaelen and Aghajanian, 1983). DRN neurons from exogenously expressing ePet-Rgs2^{hi} mice showed a higher firing frequency in contrast to Rgs2^{-/-} mice, which demonstrated a lower frequency and precision of firing compared to controls (Fig 6C-E). Moreover, exogenous expression of Rgs2 in Rgs2^{-/-} serotonergic neurons was able to recover DRN (serotonergic) neuron frequency and precision of firing to control levels (Fig 6C-E).

4. Methods: How was GCaMP6 administered? Was it specifically expressed in 5-HT neurons? Provide evidence of what proportion of infected cells were 5-HT neurons.

We include on p13 in the Methods, section **Intracranial virus injections**: GCaMP viruses were injected in anaesthetized adult mouse brains in the dorsal raphe. All mouse lines were injected with pAAV.Syn.GCaMP6f.WPRE.SV40 (Penn Vector Core, Philadelphia, PA) except for ePet-cre mice which were injected with pAAV.CAG.Flex.GCaMP6f.WPRE.SV40 (Penn Vector Core, Philadelphia, PA) for 5-HT specific neuron expression. The position of the dorsal raphe was determined according to the xyz coordination of mouse brain (AP: -4.1, ML: 0.0, DV: -1.7-2.5 mm). Mice were deeply anesthetized with 1.5-2.0% isoflurane and placed into a stereotactic frame (Narishige, Japan). The skin was opened with a sagittal incision along the midline. A small craniotomy was performed and 0.5 - 1 μ l of viruses were applied in

100 μm steps using pressure injection in 2 min intervals. A customized glass pipette attached to a 5 ml syringe was used for virus delivery. At the end of injection the skin was sutured (Surgicryl Monofilament, Belgium). After the surgery, animals received subcutaneous injection of carprofen (2 mg/kg) for analgesia. Animals were placed individually into their home cages to recover and allow for virus expression for 7-14 days.

According to Beck et al (2004)(Beck et al., 2004) 35% and 38% of electrophysiological and immunological characterized neurons within the DRN and MRN, respectively, are non serotonergic neurons. Because we did not distinguish between 5HT- and non-5HT neurons in our recordings in the manuscript of the first submission we now reanalyzed and reorganized our data according to the reviewers comments. We performed the following analysis.

1. We compared the spike rate of all analyzed GCaMP6 infected neurons in the DRN between control^{hi}, ePet-Rgs2^{hi} and Rgs2^{-/-} mice and found that the spiking rate is significantly increased for ePet-Rgs2^{hi} and significantly decreased for Rgs2^{-/-} DRN neurons when compared to control^{hi} DRN neurons (see Supplementary Fig 1D).
2. We compared the GCaMP6 signals between DRN neurons in general and 5HT neurons in particular using the recordings obtained for control^{hi} and ePet-Cre mice and blotted the spike rate against the half-width of the GCaMP signal (defined as signal duration at the halfway between baseline fluorescence and peak fluorescence). We found that in GCaMP infected DRN neurons of control^{hi} mice representing 5HT and non-5HT neurons a certain percentage of neurons reveal a broad GCaMP6 signal, which is observed to a much lesser extent in 5HT neurons from ePet-Cre mice (see Supplementary Fig 1A-C). These results suggest that a certain population of non-5HT neurons have a broad GCaMP6 signal. Under the assumption that these cells are non-5HT neurons, we excluded these cells and recalculated the spike frequencies for the remaining cells. Since 38% of neurons within the DRN/MRN are non-serotonergic neurons we used a half-width constraint of 5 sec. Using this constraint 38% of the neurons recorded in control^{hi}, 36% of Rgs2^{hi} and 35% of Rgs2^{-/-} mice were excluded. (Note, that for the 5 sec constraint only 10% of 5HT neuron from ePet-Cre mice had a larger half-width than 5). The comparison between the spike rates for width-corrected neurons reveal that overexpression of Rgs2 in 5HT neurons increases the spiking rate of 5HT neurons, while the Rgs2^{-/-} decreases the firing of 5HT neurons significantly (see Fig 8B).

We included the data on calcium spike frequency for spike width corrected experiments in Fig. 8B. The data for spike width and comparison between 5HT specific and unspecific GCaMP expression are presented in Supplementary Fig 1.

The specificity for the drug experiments are given by the cell-type specific GPCR expression and the half spike width < 5 sec.

5HT1A receptors are in particular expressed on 5HT neurons and therefore 5HT1A receptor agonist responses are either absent or reduced in non-5HT neurons in the DRN (Calizo et al., 2011). 5HT1A receptors activate the Gi/o pathway, leading to the activation of GIRK channels and to a reduction/inhibition of AP firing in 5HT neurons.

5HT1B receptors are the predominant receptor at the presynaptic terminal of 5HT neurons and are responsible for the presynaptic autoinhibition of 5-HT release in which 5-HT_{1A}-mediated slow IPSPs are reduced by previously released 5-HT activating presynaptic 5-HT_{1B} receptors (Morikawa et al., 2000).

NA causes an increase in 5-HT neuronal firing in DRN. This effect has been mainly suggested to be mediated via activation of $\alpha 1$ adrenoceptors located on 5-HT neurons coupling the the Gq/11 pathway (Day et al., 1997; Maejima et al., 2013; Vandermaelen and Aghajanian, 1983).

5. Results: Description of the transgenic lines could be moved to Methods.

On p15, in the Results under the heading "Creation of ePet-Rgs2 Transgenic Mouse Lines" we removed and added to the Methods under "Transgenic mice" on p 6: We received 9 founder lines, which were positive for the ePet-Rgs2-IRES-GFP BAC constructs. These founder lines were backcrossed for >17 generations into C57/Bl6 mice and evaluated for RGS2/GFP expression in 5HT neurons.

6. Fig. 1A: There are over 20 RGS proteins, yet the authors focus only on RGS2 and 4; it would be informative to address the relative levels of other RGS proteins in 5-HT neurons.

Based on the amino acid sequence identities within the RGS domains 5 subfamilies can be distinguished (Ross and Wilkie, 2000). RGS2 and RGS4 belong to the R4 subfamily, which contain the RGS domain and small N- and C-termini. A clear demonstration for the involvement of RGS proteins in synaptic plasticity has been shown for RGS2, RGS4, RGS7, RGS9-2 and RGS14 (Gerber et al., 2016), where RGS7, RGS9 and RGS14 belong to different RGS subfamilies. We therefore focused our study on RGS2 and RGS4, because RGS4 is the closest RGS family member involved in synaptic plasticity in the mammalian brain and may substitute for RGS2.

Using cell sorting of YFP⁺ 5HT neurons Wylie et al demonstrated the expression of 9 different RGS mRNAs in 5HT neurons in the DRN (i.e. RGS2, 3, 7, 8, 9, 10, 12, 17 and 19). In contrast RGS4, 16 and 20 are only expressed in non-5HT neurons, while RGS1, 6, 11, 13, 14, 18, 20 are not expressed at all within the DRN (Wylie et al., 2010). The differential expression of RGS2 and RGS4 in 5HT vs non-5HT neurons could be confirmed in our study (Fig 1).

We now say on p14 under the Results: ... to detect differences in *Rgs2* and the closely related RGS family member *Rgs4* mRNAs levels in serotonergic (5HT) neurons. RGS4 has also been described to modulate synaptic plasticity and to be expressed in the DRN in non-serotonergic neurons (Gerber et al., 2016; Wylie et al., 2010)

7. Fig. 2D: define the X-axis: is this the dilution factor?

On p34, Figure legend 2D we added: Serial dilutions from 2^0 to 2^4 were prepared from reversely transcribed cDNA. Single cell qRT-PCRs revealed higher *Rgs2* mRNA expression in ePet-Rgs2^{hi} compared to ePet-Rgs2^{lo} mice.

8. Fig. 4D-F: Were duration, latency or number of bites different between control and RGS2-lo or hi? Comment on why only the attack duration (and # of bites for LO) seems to show a difference from control under the RGS2 background: it suggests that RGS2 may act at other sites or developmentally to partly drive the phenotype.

Unfortunately, there were no significant differences between RGS2^{lo} and the controls. To avoid any potential background issues we used and compared littermate controls from the RGS2^{lo} and RGS2^{hi} mice instead of C57/BL6 mice for the behavior tests. Although there seems to be a slight difference (for attacks, duration, bites) between the controls lo and hi, they were not significant due to the high variability with behavior studies. These differences can be attributed to individual mice or as the reviewer suggested other sites or developmental effects where RGS2 may be driving.

9. Fig. 6B: Was there a correlation of firing rate with ISI? What is r2 and significance? Is this correlation meaningful?

The former plot in Fig 6B was to show that the DRN neurons of Rgs2^{-/-} mice have low firing rates and larger CV2s. Since this is already shown in the bar graphs of the firing rate and ISI CV2, we deleted Fig 6B.

10. Fig. 6C-E: Were the Firing rate, ISI different between RGS2hi vs. RGS2hi rescue and any other groups? In particular, RGS^{-/-} rescue should be compared with RGS^{-/-} to show a rescue.

We now included the statistical significance for Rgs^{-/-} rescue in comparison to the Rgs^{-/-}. The rescue reveals a significant increase in the firing rate, ISI CV and ISI CV2.

11. Fig. 7: Was cFos increased in 5-HT, GABA or both cell type in DRN? Co-staining with TPH or 5-HT could address this. As a control for VMHvl, quantify cFos+ cells in another adjacent part of the hypothalamus.

In Figure 7 we show increases in c-fos staining in both 5HT and GABA neurons.

As a control for VMHvl we counted the number of c-fos positive cells in the dorsomedial part of the ventromedial hypothalamus but did not detect any significant differences in c-fos expression between the mouse lines.

12. Fig. 8: How were Ca spikes detected? How were cells labeled with GCamp6, what virus, how, which cells? Ca²⁺ spikes were detected and analyzed using dedicated software written in Matlab. The fluorescence trace was calculated by measuring the mean fluorescence intensity of the region of interest per frame. The measured fluorescent intensity was corrected for a baseline value defined as the lowest (<10%) value of the fluorescence measured during the experiment. Ca²⁺ spikes were analyzed off line and were defined as spikes with a clear on- and offset. The half-width of the spikes were calculated at the median of the spike amplitude.

We included the information on Ca²⁺ spike detection in the Methods: Ca²⁺ spikes were detected and analyzed using dedicated software written in Matlab. The fluorescence trace was calculated by measuring the mean fluorescence intensity of the region of interest per frame. Regions of interest were marked before calculating $\Delta F/F$. $\Delta F/F = (F - F_0)/F_0$, where F is the fluorescence and F₀ is the baseline fluorescence. The measured fluorescent intensity was corrected for a baseline value defined as the lowest (<10%) value of the fluorescence measured during the experiment. Ca²⁺ spikes were analyzed off line and were defined as spikes with a clear on- and offset. The half-width of the spikes were calculated at the median of the spike amplitude.

We include on p13 in the Methods, section **Intracranial virus injections**: GCaMP viruses were injected in anaesthetized adult mouse brains in the dorsal raphe. All mouse lines were injected with pAAV.Syn.GCaMP6f.WPRE.SV40 (Penn Vector Core, Philadelphia, PA) except for ePet-cre mice which were injected with pAAV.CAG.Flex.GCaMP6f.WPRE.SV40 (Penn Vector Core, Philadelphia, PA) for 5-HT specific neuron expression. The position of the dorsal raphe was determined according to the xyz coordination of mouse brain (AP: -4.1, ML: 0.0, DV: -1.7-2.5 mm). Mice were deeply anesthetized with 1.5-2.0% isoflurane and placed into a stereotactic frame (Narishige, Japan). The skin was opened with a sagittal incision along the midline. A small craniotomy was performed and 0.5 - 1 μ l of viruses were applied in 100 μ m steps using pressure injection in 2 min intervals. A customized glass pipette attached to a 5 ml syringe was used for virus delivery. At the end of injection the skin was sutured (Surgicryl Monofilament, Belgium). After the surgery, animals received subcutaneous injection of carprofen (2 mg/kg) for analgesia. Animals were placed individually into their home cages to recover and allow for virus expression for 7-14 days.

For distinguishing cell-types please see above point 4.

13. Given the large number of RGS proteins, specific inhibition of RGS2 may not have a pronounced effect on 5-HT neurons due to compensation by other RGS proteins. Although changes were seen in the RGS2 knockout, these changes could involve altered activity of inputs to 5-HT neurons, rather than in 5-HT neurons. The authors could perhaps address the role of endogenous RGS2 using a viral knockdown strategy. This would be important given their main conclusion that targeting RGS2 in 5-HT neurons could be a therapeutic target.

Although an excellent suggestion, we would not be able to accomplish this in 3 months. These experiments would require another 2-3 years to perform and is an excellent follow up project. In addition our department is not setup for screening shRNAs or mRNAi for knockdown experiments and would have to establish the technique. Another huge pitfall in this field is that there are no specific antibodies available for RGS2 and other family members (see supplemental material in our publication (Han et al., 2006). Therefore we would not be able to verify the down regulation of RGS2 protein, which is critical for RGS2 knockdown experiments or that the other RGS family members were not nonspecifically down regulated at the protein level.

14. Discussion: In contrast to the model suggested for 5-HT_{1B} autoreceptor signaling to reduce 5-HT release in the raphe, enhance raphe firing and induce aggression, more recent work indicates that knockout 5-HT_{1B} receptors in the forebrain mediate aggressivity, while knockout of 5-HT_{1B} autoreceptors have an anti-anxiety or anti-depressant effect (Nautiyal et al., 2015; Nautiyal et al., 2016). Thus the effect of RGS2 on aggressivity may not be driven by inhibition of 5-HT_{1B} autoreceptors.

We agree with the reviewer and now say on p22 in the Discussion: Based on our previous study we hypothesize that 5HT release in serotonergic neurons is increased and involve the RGS2 mediated attenuation of 5HT_{1B} receptor responses . However, this has to be demonstrated in *in vivo* recordings from behaving animals and has to be compared to studies of 5HT_{1B} autoreceptor knock-out mice, which reveal an anti-anxiety and anti-depressant

phenotype rather than an aggressive phenotype, which is observed in 5HT_{1B} forebrain knockout mice (Nautiyal et al., 2015, 2016).

In addition, we expanded in the discussion (p22-23) on the specificity and modulation of RGS2 on G protein pathways in general and for α_1 adrenergic receptors in particular: RGS2 has been shown to specifically interact with α_1 adrenergic receptors via sphinophilin. This RGS2/spinophilin interaction drastically attenuate/block adrenaline induced α_1 adrenergic receptor signaling in a concentration-dependent manner, which is not observed in cells from Rgs2^{-/-} mice⁷⁰.

RGS proteins act as GTPase activating proteins, which accelerate the hydrolysis of bound GTP on the G protein α subunit. Physiologically this leads either to the inhibition of G protein signaling or the acceleration of signal termination⁶⁶. Importantly, modulation of the GPCR signal can be GPCR/RGS specific leading to the control of a subsets of GPCR signals within a neuronal/cellular population in RGS concentration-dependent manner⁷¹. Since Rgs2 is an immediate early gene, which is dynamically upregulated during activity and targets specific GPCRs and signaling pathways, one can hypothesize that the upregulation of RGS2 in serotonin neurons for example during stress will shift the serotonergic modulation in the brain to higher aggression.

Reviewer #3 (Remarks to the Author):

The current MS of Mark et al., presents the results of a series of multidisciplinary experiments aimed to test the hypothesis that the regulator of G protein signaling 2 protein (RGS2) in DRN serotonergic neurons is causally involved in regulating anxious and aggressive behavior in mice. By employing state-of-the-art genetic manipulation, behavioral monitoring and neurophysiological recording techniques, they convincingly demonstrate that overexpressing of RGS2 exclusively in DRN serotonergic neurons enhanced aggressive but not anxious behavior. Moreover, they were also able to rescue the docile phenotype of RGS2 knock-out mice by overexpressing RGS2 in serotonergic neurons. In addition, they nicely showed that the enhanced aggressive behavior in DRN RGS2 overexpressing mice is associated with enhanced neuronal activation (as assessed by the surrogate IEG marker c-fos) in dorsal raphe nuclei as well in the ventrolateral part of the VMH (shown previously as neuronal hotspot driving aggression), increased firing rate of serotonergic neurons and reduction of the modulatory actions of Gi/0 and Gq/11 –coupled 5-HT_{1B} autoreceptor and α_1 adrenergic receptors in serotonergic neurons. Clearly, given the long-term link between the CNS serotonergic system and aggression/anxiety in both humans and animals, the objectives and rationale of this study is valid and comprehensibly described in the introduction. The aim and hypothesis of the study is lucid and to the point. As already mentioned, the employed methodology is contemporary and state-of-the-art using validated techniques that are meticulously conducted. The obtained results are interesting, novel and clearly presented in both text and graphs/tables. The discussion of and conclusions drawn from the results are similarly relevant and valid. In Toto, a very nice study with that is of great interest to the field. I have only a couple of points that may be addressed in the discussion.

- 1) How do the authors explain the lack of desensitisation of the 8-OHDPAT-induced decrease in calcium spike frequency in the DRN? The somatodendritic 5-HT_{1A} autoreceptors are coupled to the Gi/o signaling pathway, just like the terminal 5-HT_{1B} autoreceptors. Given the observed increased spiking frequency of 5-HT neurons in the RGS2 overexpressor mice, one would expect this 5-HT_{1A} autoreceptor that importantly controls 5-HT neuronal activity to be desensitized.

The effects we observe do not exclude desensitization of 5HT_{1A} receptors. They only reveal that RGS2 does not attenuate 5HT_{1A} receptor responses, but do not give an estimate about the amount of functional 5HT_{1A} receptors expressed at the plasma membrane of 5HT neurons.

That RGS2 does not modulate 5HT_{1A} receptors has been shown also in other studies. For example RGS2 has little effect on 5-HT_{1A} receptor signaling in cell lines expressing 5-HT_{1A} receptors and *in vivo*, while RGS4 attenuates 5-HT_{1A} receptor responses (Beyer et al., 2004; Ghavami et al., 2004; Jaen et al., 2005). In addition, RGS6 specifically attenuates Gi/o mediated 5HT_{1A} receptors responses. Interestingly, RGS6 (-/-) reveal anxiolytic and antidepressant behavior, which is reversed by 5HT_{1A} receptor antagonists (Stewart et al., 2014).

- 2) The discussion section on the modulation of serotonergic activity and aggression states a lot of findings from other studies but no references are being given. These should be cited.

Sorry for this. For some reason the citation in the submitted version got deleted. We now included these citations on p23-24.

- 3) In the section on the neuronal activity in the DRN it is stated that the majority of increased c-fos staining was found in non-serotonergic neurons. However, given the presented results in figure 7 this does not seem to be the case as no double-staining with Tph or another specific serotonin cell marker has been performed. Hence, this claim should be removed or in case double-staining has been performed the results should be included. This aspect is particularly important as it may contribute to the somewhat dogmatic view that serotonergic neuronal activity becomes suppressed during the execution of aggressive behaviors. There is substantial (but poorly cited) evidence in the literature that the actual display of aggressive behavior is associated with robust but perhaps short-lasting increased 5-HT neuronal activity measured via different techniques (Van der Vegt et al., 2003; Veening et al., 2005; Haller et al., 2005; 2006; Ferris et al., 2008; Terranova et al., 2016; Cadogan et al., 1994; Takahashi et al., 2011; 2015; Nakazato 2013; Agoy et al., 2013).

We agree with the reviewer and removed the claim that the majority of increased c-fos staining was found in non-serotonergic neurons, since we have not performed a complete study on the co-staining. We also included now in the discussion most of the suggested references and reorganized the Discussion accordingly on p23-24: In contrast, during the performance and expression of aggressive behavior 5HT levels have been shown to be increased (Takahashi et al., 2011). For example activation of GABA_A or 5HT_{1A} receptors in the DRN, which leads to the inhibition of 5HT neuronal activity, decreases aggressive behaviors and high-aggressive vs low aggressive rats reveal an increased number of c-fos

positive 5HT neurons in the DRN and MRN categorized by their performance in the residence intruder test (van der Vegt et al., 2003). Rats and mice exposed to aggressive and agonistic encounters also reveal an increase in c-fos positive 5HT neurons or DRN neurons (Haller et al., 2005, 2006). In addition, acquisition of dominance behavior in female hamsters has been associated with activation of DRN 5HT neurons and 5HT1A agonists injected into the hypothalamus or systemically administered fluoxetine increases aggression (note, that male hamsters reveal the opposite behaviors) (Terranova et al., 2016). Social interaction with an unfamiliar partner and area also results in an increase in extracellular 5HT levels in the ventral hippocampus (Cadogan et al., 1994).

Added to Discussion on p25: Upregulation of 5HT neuronal activity during aggression may involve glutamatergic and GABAergic input into the DRN and recruitment of local feedforward and feedback circuits. For example increased glutamate release causes a phasic increase of 5HT-release from DRN 5-HT neurons and is observed during escalated aggression. The process may involve activation of GABA_B receptors localized on non-serotonergic neurons in the DRN, which leads to an increase in glutamate release (Takahashi et al., 2015).

REVIEWERS' COMMENTS:

Reviewer #1 (Remarks to the Author):

With this resubmission, the authors have greatly improved upon their initial manuscript. Mark et al. have addressed all of this reviewer's comments satisfactorily, through the expanded analysis of their data and the inclusion of an expanded Discussion section. I have no further comments or suggestions to add.

Reviewer #2 (Remarks to the Author):

The authors have largely addressed my previous comments, with a careful analysis of the firing and calcium responses of 5-HT vs. non-5-HT neurons of DRN. Although they did not specifically knock down RGS2 in 5-HT neurons, the manuscript presents important new data on the role of RGS2 overexpression in 5-HT function.

A few minor issues remain.

Specific comments:

1. Rebuttal: The correction of DRN coordinates "-4.24-4.9, ML: \pm 0.25, DV: -1800-3000 μ M" does not make sense: use consistent units (mm or μ m, nor μ M). However in the paper, it seems that the correct coordinates are given on p. 11; please confirm.
2. Fig. 8B/Supp Fig 1: The authors excluded non-5-HT neurons based on half-width constraint of GCaMP signal of >5 sec; Were there any changes in spiking rate in the non-5-HT neurons in RGS2 over-expressing or RGS2^{-/-} samples?
3. References: Several of the references lack volume and page numbers and need revision.
4. Discussion: it seems possible that RGS2 inhibition of 5-HT_{1B}-mediated presynaptic inhibition may be important to drive the aggression phenotype in RGS2 over-expression, by leading to activation of 5-HT release. It is unclear why in RGS2^{-/-} mice anxiety/depression phenotype as seen in 5-HT_{1B} autoreceptor ^{-/-} mice (Nautiyal), but the latter may be due to a developmental effect, since the 5-HT_{1B} recovered in adulthood in this model.

Reviewer #3 (Remarks to the Author):

The authors have satisfactorily addressed all my points and the current revision has improved. The MS will be a nice addition to the research field.

Reviewer #1 (Remarks to the Author):

With this resubmission, the authors have greatly improved upon their initial manuscript. Mark et al. have addressed all of this reviewer's comments satisfactorily, through the expanded analysis of their data and the inclusion of an expanded Discussion section. I have no further comments or suggestions to add.

Reviewer #2 (Remarks to the Author):

The authors have largely addressed my previous comments, with a careful analysis of the firing and calcium responses of 5-HT vs. non-5-HT neurons of DRN. Although they did not specifically knock down RGS2 in 5-HT neurons, the manuscript presents important new data on the role of RGS2 overexpression in 5-HT function.

A few minor issues remain.

Specific comments:

1. Rebuttal: The correction of DRN coordinates “-4.24-4.9, ML: \pm 0.25, DV: -1800-3000 μ M” does not make sense: use consistent units (mm or μ m, nor μ M). However in the paper, it seems that the correct coordinates are given on p. 11; please confirm.

The coordinates used for the DRN on p11 and p13 are correct using mm as units.

2. Fig. 8B/Supp Fig 1: The authors excluded non-5-HT neurons based on half-width constraint of GCaMP signal of >5 sec; Were there any changes in spiking rate in the non-5-HT neurons in RGS2 over-expressing or RGS2^{-/-} samples?

There were no significant changes in the spiking frequencies for half-width constraint of GCaMP signals of >5 sec between control^{hi} and Rgs2^{-/-} neurons (control^{hi} 0.026 \pm 0.003 Hz (n=59); Rgs2^{-/-} 0.025 \pm 0.004 Hz (n=36), p > 0.38). However, there was a significant increase in the firing frequency of GCaMP signals of >5 sec for ePet^{Rgs2hi} neurons vs control^{hi} and Rgs2^{-/-} neurons (0.037 \pm 0.004 Hz (n=44); p<0.05). These results may suggest an upregulation of the activity of non-5HT neurons in the DRN in high aggressive mice (see ref 82 and discussion).

3. References: Several of the references lack volume and page numbers and need revision.

References

1. Changed yr 2007 to vol **370**
10. Added **168B**, 211-222 and deleted DOI
11. Added **26**, 309-315 and deleted DOI
12. Added **113**, 1921-1925
14. Added **65**, 298-308 and deleted DOI
15. Added **39**, 1340-1346
16. Added **23**, 369-373 and deleted DOI
18. Corrected authors Gottschalk M. G. & Domschke K.
22. Added **1453**, 26-33 and deleted DOI
23. Added **53**, 61-82 and deleted DOI
27. Deleted an A. and M. in author initials

30. Added 37807-14
33. Deleted M. in first author initials
34. Deleted M. in first author initials
36. Added **13**, 167-170 and deleted DOI
38. Added **316**, R1-2 and deleted DOI
41. Added **153**, 169-175 and deleted DOI
45. Added **89**, 273-86 and deleted DOI
48. Added **20**, 295-302 and deleted DOI
51. Added **10**, 195 and deleted DOI
52. Added **19**, 596-604 and deleted DOI
53. Added **470**, 221-226 and deleted DOI
57. Added **89**, 273-286 and deleted DOI
58. Added **17**, e12420 and deleted DOI
- 59&60 Corrected authors initials
61. Added **582**, 481-488 and deleted DOI
62. Added **23**, 318-324 and deleted DOI
63. Added **24**, 348-360 and deleted DOI
66. Added **18**, R777-R783 and deleted DOI
67. Added **86**, 813-826 and deleted DOI
68. Added **41**, 2941-2950 and deleted DOI
71. Added **366**, 349-365 and deleted DOI
76. Added **161**, 88-94 and deleted DOI
77. Added **88**, 173-182 and deleted DOI
78. Added **113**, 13233-13238 and deleted DOI
79. Added **5**, 299-305 and deleted DOI
81. Added **10**, 20-33 and deleted DOI
82. Added **35**, 6452-63 and deleted DOI
83. Added 43

4. Discussion: it seems possible that RGS2 inhibition of 5-HT_{1B}-mediated presynaptic inhibition may be important to drive the aggression phenotype in RGS2 over-expression, by leading to activation of 5-HT release. It is unclear why in RGS^{hi} is no reduced anxiety/depression phenotype as seen in 5-HT_{1B} autoreceptor ^{-/-} mice (Nautiyal), but the latter may be due to a developmental effect, since the 5-HT_{1B} recovered in adulthood in this model.

We agree with the reviewer that the anxiety/depression phenotype may be due to a developmental effect or due to other affected areas of the brain.

Reviewer #3 (Remarks to the Author):

The authors have satisfactorily addressed all my points and the current revision has improved. The MS will be a nice addition to the research field.